# SARS-CoV-2 uses CD4 to infect T helper lymphocytes

Natalia S Brunetti[1†], Gustavo G Davanzo[2†], Diogo de Moraes[3,4†], Allan JR Ferrari[5†], Gabriela F Souza[6†], Stéfanie Primon Muraro[6†], Thiago L Knittel[3†], Vinicius O Boldrini[1†], Lauar B Monteiro[2†], João Victor Virgílio-da-Silva[2†], Gerson S Profeta[3‡], Natália S Wassano[7‡], Luana Nunes Santos[8‡], Victor C Carregari[9‡], Artur HS Dias[5‡], Flavio P Veras[10,11,12], Lucas A Tavares[13], Julia Forato[6], Icaro MS Castro[14], Lícia C Silva-Costa[9], André C Palma[15], Eli Mansour[15], Raisa G Ulaf[15], Ana F Bernardes[15], Thyago A Nunes[15], Luciana C Ribeiro[15], Marcus V Agrela[15], Maria Luiza Moretti[15], Lucas I Buscaratti[8], Fernanda Crunfli[9], Raissa G Ludwig[3], Jaqueline A Gerhardt[7], Natália Munhoz-Alves[1], Ana Maria Marques[1], Renata Sesti-Costa[1,16], Mariene R Amorim[6], Daniel A Toledo-Teixeira[6], Pierina Lorencini Parise[6], Matheus Cavalheiro Martini[6], Karina Bispos-dos-Santos[6], Camila L Simeoni[6], Fabiana Granja[6], Virgínia C Silvestrini[17], Eduardo B de Oliveira[17], Vitor M Faca[17], Murilo Carvalho[18,19], Bianca G Castelucci[18,19], Alexandre B Pereira[18], Laís D Coimbra[18], Marieli MG Dias[18], Patricia B Rodrigues[20], Arilson Bernardo SP Gomes[20], Fabricio B Pereira[16], Leonilda MB Santos[21,22], Louis-Marie Bloyet[23], Spencer Stumpf[23], Marjorie C Pontelli[23], Sean Whelan[23], Andrei C Sposito[24], Robson F Carvalho[4], André S Vieira[25], Marco AR Vinolo[20,26,27], André Damasio[7,26], Licio Velloso[15,27], Ana Carolina M Figueira[18], Luis LP da Silva[10], Thiago Mattar Cunha[10,12], Helder I Nakaya[14], Henrique Marques-Souza[7,8], Rafael E Marques[18], Daniel Martins-de-Souza[9,26,28,29], Munir S Skaf[5], Jose Luiz Proenca-Modena[6,26], Pedro MM Moraes-Vieira[2,26,27], Marcelo A Mori[3,26,27]*, Alessandro S Farias[1,23,26,27]*

*For correspondence: morima@unicamp.br (MAM); asfarias@unicamp.br (ASF)

†These authors contributed equally to this work
‡These authors also contributed equally to this work

**Competing interest:** The authors declare that no competing interests exist.

[1]Autoimmune Research Laboratory, Department of Genetics, Microbiology and Immunology, Institute of Biology, University of Campinas (UNICAMP), Campinas, Brazil; [2]Laboratory of Immunometabolism, Department of Genetics, Microbiology and Immunology, Institute of Biology, University of Campinas (UNICAMP), Campinas, Brazil; [3]Laboratory of Aging Biology, Department of Biochemistry and Tissue Biology, Institute of Biology, University of Campinas (UNICAMP), Campinas, Brazil; [4]Department of Structural and Functional Biology, Institute of Biosciences, Sao Paulo State University (UNESP), Botucatu, Brazil; [5]Institute of Chemistry and Center for Computing in Engineering and Sciences, University of Campinas, Campinas (UNICAMP), Campinas, Brazil; [6]Laboratory of Emerging Viruses, Department of Genetics, Microbiology and Immunology, Institute of Biology, University of Campinas (UNICAMP), Campinas, Brazil; [7]Department of Biochemistry and Tissue Biology, Institute of Biology, University of Campinas (UNICAMP), Campinas, Brazil; [8]Brazilian Laboratory on Silencing Technologies (BLaST), Department of Biochemistry and Tissue Biology, Institute of Biology, University of Campinas (UNICAMP), Campinas, Brazil; [9]Laboratory of Neuroproteomics, Department of Biochemistry and Tissue Biology, Institute of Biology, University of Campinas (UNICAMP), Campinas, Brazil; [10]Center of Research in Inflammatory Diseases, Ribeirão Preto Medical School, University of São Paulo, Ribeirão Preto, São Paulo, Brazil; [11]Department of BioMolecular Sciences, School of Pharmaceutical Sciences, University of São

Paulo, Ribeirão Preto, São Paulo, Brazil; [12]Department of Pharmacology, Ribeirão Preto Medical School, University of São Paulo, Ribeirão Preto,, São Paulo, Brazil; [13]Department of Cell and Molecular Biology, Center for Virology Research, Ribeirão Preto Medical School, University of São Paulo, Ribeirão Preto, Brazil; [14]Department of Clinical and Toxicological Analyses, School of Pharmaceutical Sciences, University of São Paulo, São Paulo, Brazil; [15]Department of Internal Medicine, School of Medical Sciences, University of Campinas (UNICAMP), Campinas, Brazil; [16]Hematology and Hemotherapy Center, University of Campinas (UNICAMP), Campinas, Brazil; [17]Department of Biochemistry and Immunology, Ribeirão Preto Medical School, University of São Paulo, Ribeirão Preto, Brazil; [18]National Biosciences Laboratory (LNBio), Brazilian Center for Research in Energy and Materials (CNPEM), Campinas, Brazil; [19]Brazilian Synchrotron Light Laboratory (LNLS), Brazilian Center for Research in Energy and Materials (CNPEM), Campinas, Brazil; [20]Laboratory of Immunoinflammation, Department of Genetics, Microbiology and Immunology, Institute of Biology, University of Campinas (UNICAMP), Campinas, Brazil;, Campinas, Brazil; [21]Neuroimmunology Unit, Department of Genetics, Microbiology and Immunology, University of Campinas (UNICAMP), Campinas, Brazil; [22]National Institute of Science and Technology on Neuroimmunomodulation (INCT-NIM) – Oswaldo Cruz Institute, Oswaldo Cruz Foundation, Rio de Janeiro, Brazil; [23]Washington University in St Louis, Department of Molecular Microbiology, St. Louis, United States; [24]Laboratory of Vascular Biology and Arteriosclerosis, School of Medical Sciences, University of Campinas (UNICAMP), Campinas, Brazil; [25]Laboratory of Electrophysiology, Neurobiology and Behavior, University of Campinas (UNICAMP), Campinas, Brazil; [26]Experimental Medicine Research Cluster (EMRC), University of Campinas (UNICAMP), Campinas, Brazil; [27]Obesity and Comorbidities Research Center (OCRC), University of Campinas (UNICAMP), Campinas, Brazil; [28]D'Or Institute for Research and Education (IDOR), São Paulo, Brazil; [29]National Institute of Science and Technology in Biomarkers for Neuropsychiatry (INCTINBION), São Paulo, Brazil

**Abstract** The severe acute respiratory syndrome coronavirus 2 (SARS-CoV-2) is the agent of a major global outbreak of respiratory tract disease known as Coronavirus Disease 2019 (COVID-19). SARS-CoV-2 infects mainly lungs and may cause several immune-related complications, such as lymphocytopenia and cytokine storm, which are associated with the severity of the disease and predict mortality. The mechanism by which SARS-CoV-2 infection may result in immune system dysfunction is still not fully understood. Here, we show that SARS-CoV-2 infects human CD4+ T helper cells, but not CD8+ T cells, and is present in blood and bronchoalveolar lavage T helper cells of severe COVID-19 patients. We demonstrated that SARS-CoV-2 spike glycoprotein (S) directly binds to the CD4 molecule, which in turn mediates the entry of SARS-CoV-2 in T helper cells. This leads to impaired CD4 T cell function and may cause cell death. SARS-CoV-2-infected T helper cells express higher levels of IL-10, which is associated with viral persistence and disease severity. Thus, CD4-mediated SARS-CoV-2 infection of T helper cells may contribute to a poor immune response in COVID-19 patients.

## Editor's evaluation

This study represents an important contribution where SARS-CoV-2 infection of T-helper cells is implicated and found to be mediated by CD4. It identifies the interaction between spike RBD domain and N Terminal domain of CD4 molecule as the specific viral attachment strategy. This solid paper also provides a potential usefulness for future work in understanding how viruses mediate infection of T cells.

## Introduction

Coronavirus Disease 2019 (COVID-19) has rapidly spread across the globe, being declared a pandemic by the World Health Organization on March 11, 2020. COVID-19 has caused millions of deaths around the world. Most of the deaths are associated with acute pneumonia, cardiovascular complications, and organ failure due to hypoxia, exacerbated inflammatory responses, and widespread cell death (*Candido et al., 2020*; *Laing et al., 2020*; *Li et al., 2020*; *Moore and June, 2020*; *Zhang et al., 2020*). Individuals that progress to the severe stages of COVID-19 manifest marked alterations in the immune response, characterized by reduced overall protein synthesis, cytokine storm, lymphocytopenia, and T cell exhaustion (*Arunachalam et al., 2020*; *De Biasi et al., 2020*; *Moore and June, 2020*; *Thoms et al., 2020*). In addition to these acute effects on the immune system, a considerable proportion of infected individuals present low titers of neutralizing antibodies (*Robbiani et al., 2020*; *Souza et al., 2021*). Moreover, the levels of antibodies against severe acute respiratory syndrome coronavirus 2 (SARS-CoV-2) decay rapidly after recovery in part of the infected individuals (*Long et al., 2020*), suggesting that SARS-CoV-2 infection may exert profound and long-lasting complications to adaptive immunity. Recently, it has been shown that SARS-CoV-2 is able to infect lymphocytes (*Pontelli et al., 2022*; *Shen et al., 2022*). In this context, it is urgent to characterize the replicative capacity and the effects of SARS-CoV-2 replication in different immune cells, especially those involved with the formation of immunological memory and effective adaptive response, such as CD4$^+$ T lymphocytes.

In what has been proposed to be the canonical mechanism of SARS-CoV-2 infection, the spike glycoprotein of SARS-CoV-2 (sCoV-2) binds to the host angiotensin-converting enzyme 2 (ACE2), after which it is cleaved by TMPRSS2 (*Hoffmann et al., 2020*). While TMPRSS2 is ubiquitously expressed in human tissues (*Figure 1—figure supplement 1*), ACE2 is mainly expressed in epithelial and endothelial cells, as well as in the kidney, testis, and small intestine (*Figure 1—figure supplement 1*). Still, a wide variety of cell types are potentially infected by SARS-CoV-2 (*Codo et al., 2020*; *Crunfli et al., 2022*; *Lamers et al., 2020*; *de Oliveira et al., 2022*; *Pascoal et al., 2021*; *Saccon et al., 2022*; *Yang et al., 2020*), even though some of these cells express very low levels of ACE2. We showed that this is the case for lymphocytes (*Figure 1—figure supplement 2*). These findings suggest that either SARS-CoV-2 uses alternative mechanisms to enter these cells or that auxiliary molecules at the plasma membrane may promote infection by stabilizing the virus until it interacts with ACE2. In agreement with the latter, binding of sCoV-2 to certain cell surface proteins facilitates viral entry (*Cantuti-Castelvetri et al., 2020*; *Cecon et al., 2022*).

Since the structures of the spike of SARS-CoV-1 (sCoV-1) and the sCoV-2 proteins are similar (*Wrapp et al., 2020*; *Yuan et al., 2020*), we used the P-HIPSTer algorithm to uncover human proteins that putatively interact with spike (*Lasso et al., 2019*). Seventy-one human proteins were predicted to interact with sCoV-1 (*Figure 1—figure supplement 3*). We then cross-referenced the proteins with five databases of plasma membrane proteins to identify the ones located on the cell surface (see 'Methods' for details). CD4 was the only protein predicted to interact with sCoV-1 that appeared in all five databases (*Figure 1—figure supplement 3*). CD4 is expressed mainly in T helper lymphocytes and has been shown to be the co-receptor to HIV (*Shaik et al., 2019*). Since CD4$^+$ T lymphocytes orchestrate innate and adaptive immune response (*Gasteiger and Rudensky, 2014*; *O'Shea and Paul, 2010*), infection of CD4$^+$ T cells by SARS-CoV-2 could explain lymphocytopenia and dysregulated inflammatory response in severe COVID-19 patients. Moreover, from an evolutionary perspective, infection of CD4$^+$ T cells represents an effective mechanism for viruses to escape the immune response (*Xu et al., 2001*).

To test whether human primary T cells are infected by SARS-CoV-2, we purified CD3$^+$CD4$^+$ and CD3$^+$CD8$^+$ T cells from the peripheral blood of noninfected healthy controls/donors (HC), incubated these cells with SARS-CoV-2 for 1 hr, and washed them the three times with PBS. The viral load was measured 24 hr post infection. We were able to detect SARS-CoV-2 RNA in CD4$^+$ T cells, but not CD8$^+$ T cells (*Figure 1A* and *Figure 1—figure supplement 4A*). To confirm the presence of SARS-CoV-2 in the cells, we performed in situ hybridization using probes against the viral RNA-dependent RNA polymerase (RdRp) gene, immunofluorescence for sCoV-2 using antibodies against spike protein and transmission electron microscopy (*Figure 1B and C*). In parallel, we infected primary CD4$^+$ T cells with the VSV-mCherry-SARS-CoV-2 pseudotype virus (*Figure 1—figure supplement 4B*). All approaches confirmed that SARS-CoV-2 infects CD4$^+$ T cells. Notably, the amount of SARS-CoV-2 RNA remains roughly stable until 48 hr post infection (*Figure 1D*). Of note, although most of the data presented

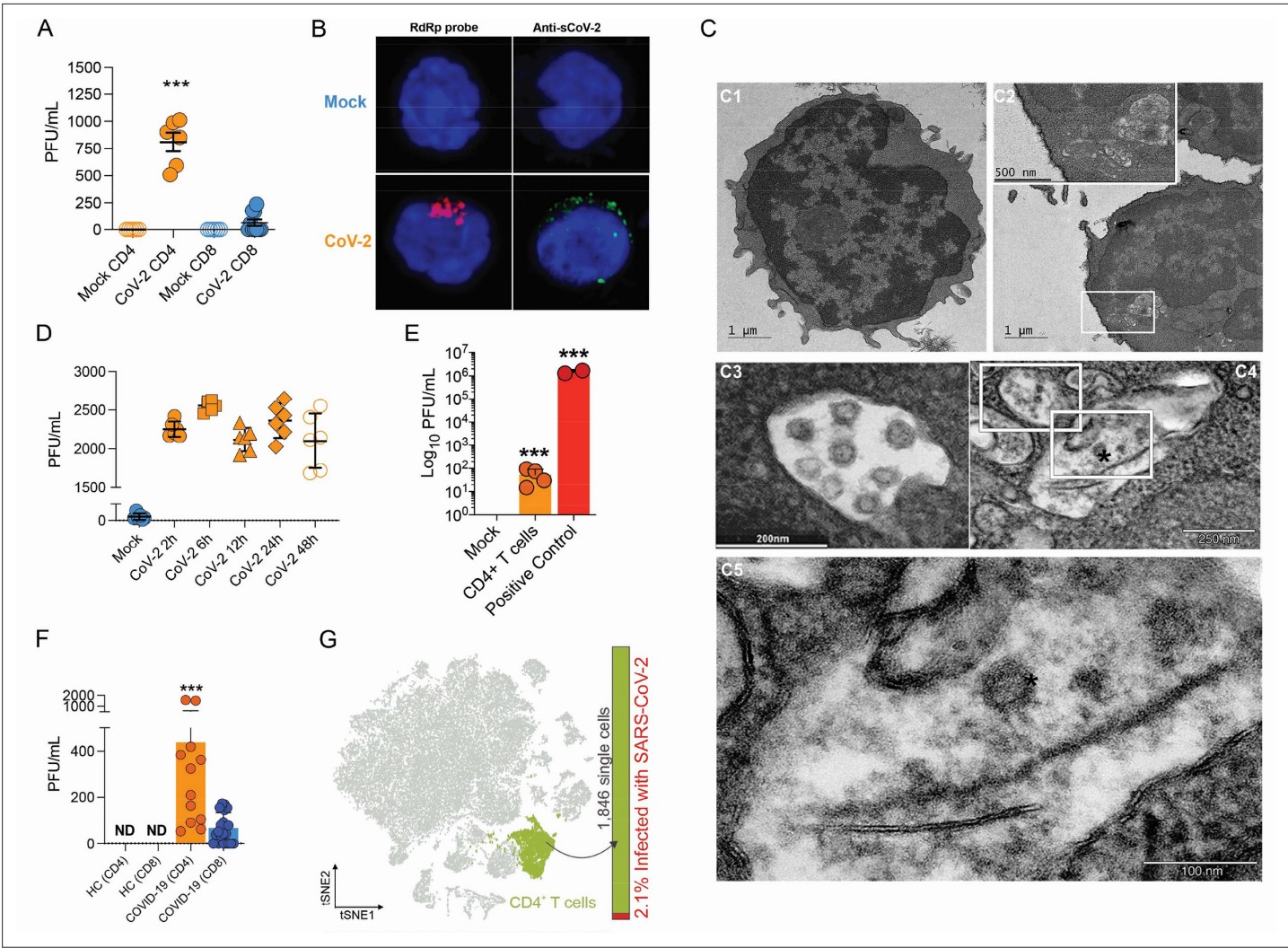

**Figure 1.** Severe acute respiratory syndrome coronavirus 2 (SARS-CoV-2) infects CD4+ T cells in vitro and in vivo. Peripheral blood CD3+CD4+ and CD3+CD8+ were exposed to mock control or SARS-CoV-2 (CoV-2) (Multiplicity of Infection (MOI) = 1) for 1 hr under continuous agitation. (**A**) Viral load was assessed by RT-qPCR 24 hr after infection. (**B**) Cells were washed, cultured for 24 hr, fixed with 4% paraformaldehyde (PFA), and stained with the RdRp probe for in situ hybridization or anti-sCoV-2 for immunofluorescence. Cells were analyzed by confocal microscopy. (**C, C1–C5**) Representative transmission electron microscopy (EM) micrographs showing viral particles (asterisks) inside lymphocytes 2 hr after infection. (**D**) Viral load was determined by qPCR 2 hr, 6 hr, 12 hr, 24 hr, and 48 hr after infection. (**E**) Vero cells were incubated with the supernatant of mock control or CoV-2-infected CD4+T cells under continuous agitation for 1 hr. The viral load in Vero cells was measured after 72 hr using plaque assay. PFU, plaque-forming unit. (**F**) Viral load was measured by RT-qPCR in peripheral blood CD4+ and CD8+ T cells from healthy controls (HC) and COVID-19 patients. (**G**) CoV-2 RNA detection in CD4+ T cells from bronchoalveolar lavage fluid (BALF) single-cell RNA sequencing data revealing the presence of CoV-2 RNA. Data represents mean ± SEM of at least two independent experiments performed in triplicate or duplicate (f). ***p<0.001; ND, not detected.

The online version of this article includes the following source data and figure supplement(s) for figure 1:

**Figure supplement 1.** Comprehensive analysis of *ACE2*, *TMPRSS2*, and *CD4* expression in human tissues and cell types.

**Figure supplement 2.** *ACE2* expression in peripheral blood leukocytes.

**Figure supplement 3.** Proteomic *in silico* analysis unveils interactions of SARS-CoV-1 with human cellular membrane proteins.

**Figure supplement 4.** Evaluation of SARS-CoV-2 infection dynamics in CD4+ T cells.

**Figure supplement 4—source data 1.** Original agarose gel with PCR for the antisense CoV-2 strand in CD4+ T and Vero (positive control) cells infected with mock or CoV-2.

**Figure supplement 5.** Gene signature of individual CD4+ T cells population.

here was generated using the ancient SARS-CoV-2 B lineage, CD4$^+$ T cells were also infected by the P.1 (gamma) variant (*Figure 1—figure supplement 4C*). Furthermore, we identified the presence of the negative strand (antisense) of SARS-CoV-2 in the infected cells (*Figure 1—figure supplement 4D*), demonstrating that the virus replicates in T helper cells. We also found by plaque assay analysis that SARS-CoV-2-infected CD4$^+$ T cells release infectious viral particles, although much less efficiently than Vero cells (positive control) (*Figure 1E* and *Figure 1—figure supplement 4E*).

To investigate whether SARS-CoV-2 infects CD4$^+$ T cells in vivo, we purified CD4$^+$ and CD8$^+$ T cells from peripheral blood cells of COVID-19 patients (*Supplementary file 1*). Similar to our ex vivo experiments, SARS-CoV-2 RNA was detected in CD4$^+$ T cells from COVID-19 patients (*Figure 1F*). Using

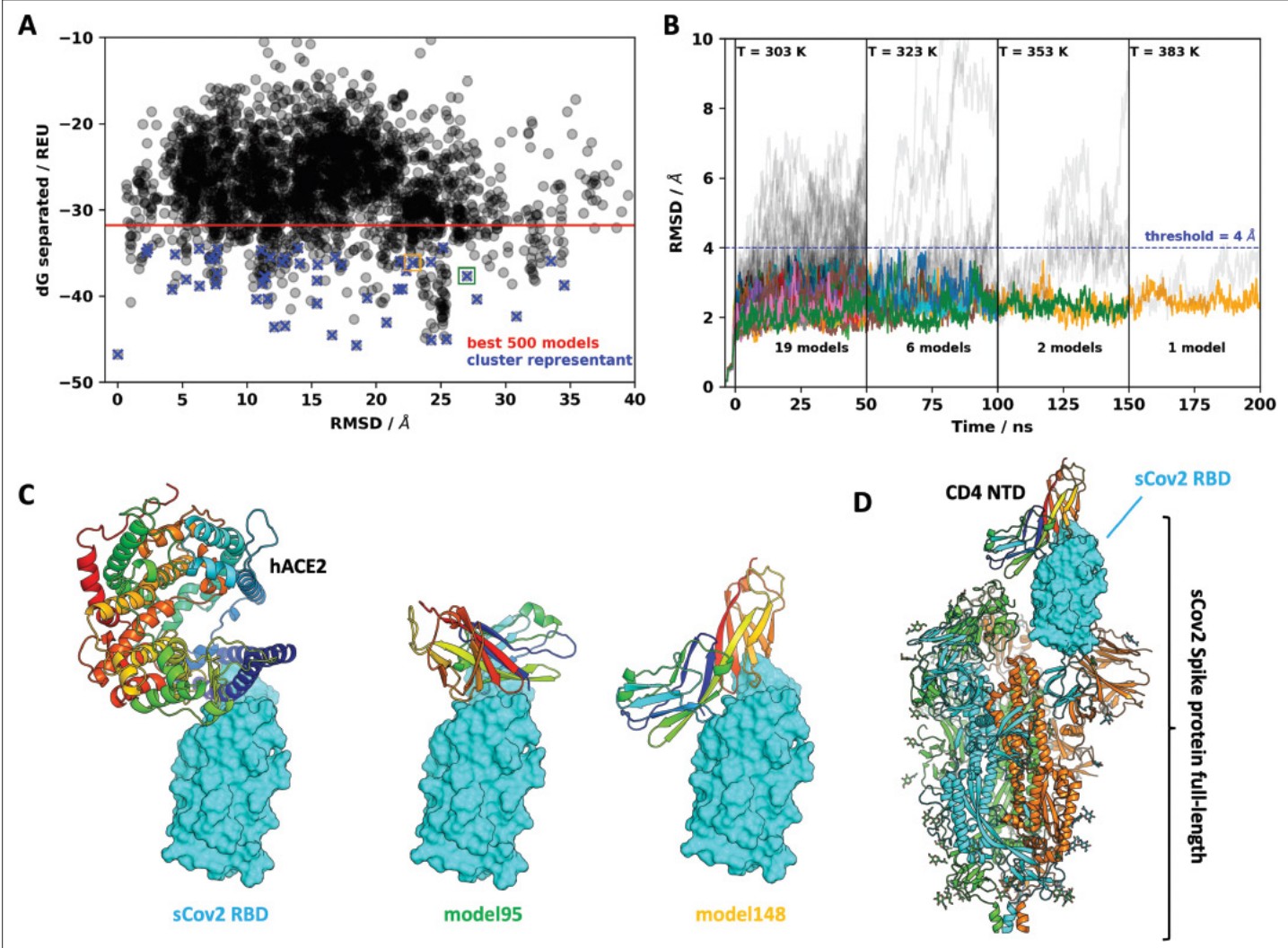

**Figure 2.** Molecular dynamics simulations of the sCoV-2 receptor binding domain (RBD) and CD4 N-terminal domain (NTD) interaction. (**A**) Interaction energy (dG separated) versus root-mean-square deviation (RMSD) plot shows a high diversity of binding modes with similar energies. Among the 500 best models, 50 cluster best-ranked representatives (shown as blue crosses) were selected for further evaluation with molecular dynamics simulations. (**B**) RMSD to the initial docked complex as a function of molecular dynamics simulation time over 200 ns and four steps of temperature. At each temperature step, well-behaved models are depicted as colored curves, while divergent candidates are shown as gray curves. Kinetically stable models making reasonable interactions remain close to their initial docked conformation. Only two models remain stable after the third step of 50 ns at 353 K. For both models, resilient contacts throughout simulation are shown in Fig. S5. (**C**) Interaction models of ACE2 and sCoV-2 RBD. The two best candidates according to molecular dynamics simulation, model 95 and model 148, present distinct binding modes. For the first case, interaction occurs mainly in the N-terminal portion of CD4 NTD, while the latter have important contributions to the central part of this domain. (**D**) Full-length model of sCoV-2 and CD4 NTD interaction obtained by alignment of sCoV-2 RBD from model 148 to the sCoV-2 RBD open state from PDB 6vyb EM structure.

The online version of this article includes the following figure supplement(s) for figure 2:

**Figure supplement 1.** Molecular dynamics simulations of the sCoV-2 receptor binding domain (RBD) and CD4 N-terminal domain (NTD) interaction.

publicly available single-cell RNA sequencing (scRNAseq) data (*Liao et al., 2020*), we detected the presence of SARS-CoV-2 RNA in 2.1% of CD4+ T cells of the bronchoalveolar lavage (BAL) of patients with severe COVID-19 (*Figure 1F* and *Figure 1—figure supplement 5*). Thus, our data demonstrate that SARS-CoV-2 infects CD4+ T cells.

Next, we sought to explore the role of the CD4 molecule in SARS-CoV-2 infection. Based on the putative interaction found using P-HIPSTer, we performed molecular docking analyses and predicted that sCoV-2 receptor binding domain (RBD) directly interacts with the N-terminal domain (NTD) of CD4 Ig-like V type (*Figure 2A* and *Figure 2—figure supplement 1*). Molecular dynamics (MD) simulations with stepwise temperature increase were applied to challenge the kinetic stability of the docking model representatives (*Figure 2B*). Two models remained stable after the third step of simulation at 353 K and represent likely candidates for the interaction between the CD4 NTD and sCoV-2 RBD (*Figure 2B*). Additionally, we evaluated the dynamic behavior of closely related binding mode models present in the same cluster as these two models. We observed a strong structural convergence toward the putative model in one case, which indicates plausible and rather stable interaction between CD4 NTD and sCoV-2 RBD (*Figure 2—figure supplement 1*). The interaction region of RBD with CD4 is predicted to overlap with that of human ACE2 (*Figure 2C and D*). The interaction between CD4 and sCoV-2 was confirmed by co-immunoprecipitation (*Figure 3A*). Binding affinity isotherms, obtained by fluorescence anisotropy assay, confirmed the physical high-affinity interaction between RBD and CD4 (Kd = 22 nM) and spike full length and CD4 (Kd = 27 nM). Considering the similar affinities, these results suggest that the interaction interface between sCoV-2 and CD4 occurs at the RBD (*Figure 3B and C*). Consistent with the hypothesis that CD4-sCoV-2 interaction is required for infection, we observed that pre-incubation with soluble CD4 (sCD4) completely blunted viral load in CD4+ T cells exposed to SARS-CoV-2 (*Figure 3D*). Together, these data demonstrate that sCoV-2 binds to the CD4 molecule.

To gain further insight into the importance of CD4-sCoV-2 binding to SARS-CoV-2 infection, we purified CD4+ T cells and preincubated them with a CD4 monoclonal antibody (RPA-T4) (*O'Shea and Paul, 2010*). We observed that CD4 inhibition reduced SARS-CoV-2 load (*Figure 3E*). Moreover, we used human T cell lines that express CD4 (A3.01) or not (A2.01) (*Folks et al., 1985*; *Figure 3—figure supplement 1*). Despite expressing higher levels of ACE2 and TMPRSS2 than primary CD4+ T cells (*Figure 1—figure supplement 2* and *Figure 3—figure supplement 1*), the presence of CD4 was sufficient to increase viral load in A3.01 cells compared to A2.01 (*Figure 3—figure supplement 1*). These results were confirmed by using the VSV-mCherry-SARS-CoV-2 pseudotype model (*Figure 3F*). Importantly, the introduction of CD4 in A2.01 increased viral load (*Figure 3H and H*).

Our immunoprecipitation experiments indicated no physical interaction between CD4 and ACE2 (data not shown). Since CD4+ T cells have very low *ACE2* expression, we tested whether CD4 alone was sufficient to allow SARS-CoV-2 entry. Inhibition of ACE2 using polyclonal antibody (*Figure 3I*) diminished SARS-CoV-2 entry in CD4+ T cells. Moreover, the inhibition of TMPRSS2 with camostat mesylate reduced SARS-CoV-2 load (*Figure 3H*). Altogether, these data demonstrate that ACE2, TMPRSS2, and CD4 are all required to allow the infection of CD4+ T cells by SARS-CoV-2.

To assess the consequences of SARS-CoV-2 infecting CD4+ T cells, we performed mass spectrometry-based shotgun proteomics in CD4+ T cells exposed to SARS-CoV-2 ex vivo. We found that SARS-CoV-2 infection alters multiple housekeeping pathways associated with the immune system, infectious diseases, cell cycle, and cellular metabolism (*Figure 4A and B* and *Figure 4—figure supplements 1 and 2*). SARS-CoV-2 exposure elicits alterations associated with 'cellular responses to stress,' which include changes in proteins involved in translation, mitochondrial metabolism, cytoskeleton remodeling, cellular senescence, and apoptosis (*Figure 4B* and *Supplementary file 2*). In agreement, ex vivo exposure of CD4+ T cells with SARS-CoV-2 resulted in 10% reduction of cell viability 24 hr after infection with a low MOI (0.1) (*Figure 4—figure supplement 3*). SARS-CoV-2 may cause cell death by different mechanisms. SARS-CoV-2 can induce apoptosis in different cell lines through stimulation of autophagy via inhibition of mTOR signaling (*Li et al., 2021*). It can also induce inflammasome activation and pyroptosis in primary monocytes and peripheral blood mononuclear cells (PBMCs) from COVID-19 patients (*Alvim et al., 2020*). In epithelial respiratory and kidney cell lines, SARS-CoV drives multimodal necrosis (*Yue et al., 2018*). In lymphocytes, however, it has been demonstrated that apoptosis is the mechanism underlying SARS-CoV-2-mediated cell death (*Crunfli et al., 2022*), which is in line with our proteomic data (*Figure 4—figure supplements 1 and 2*).

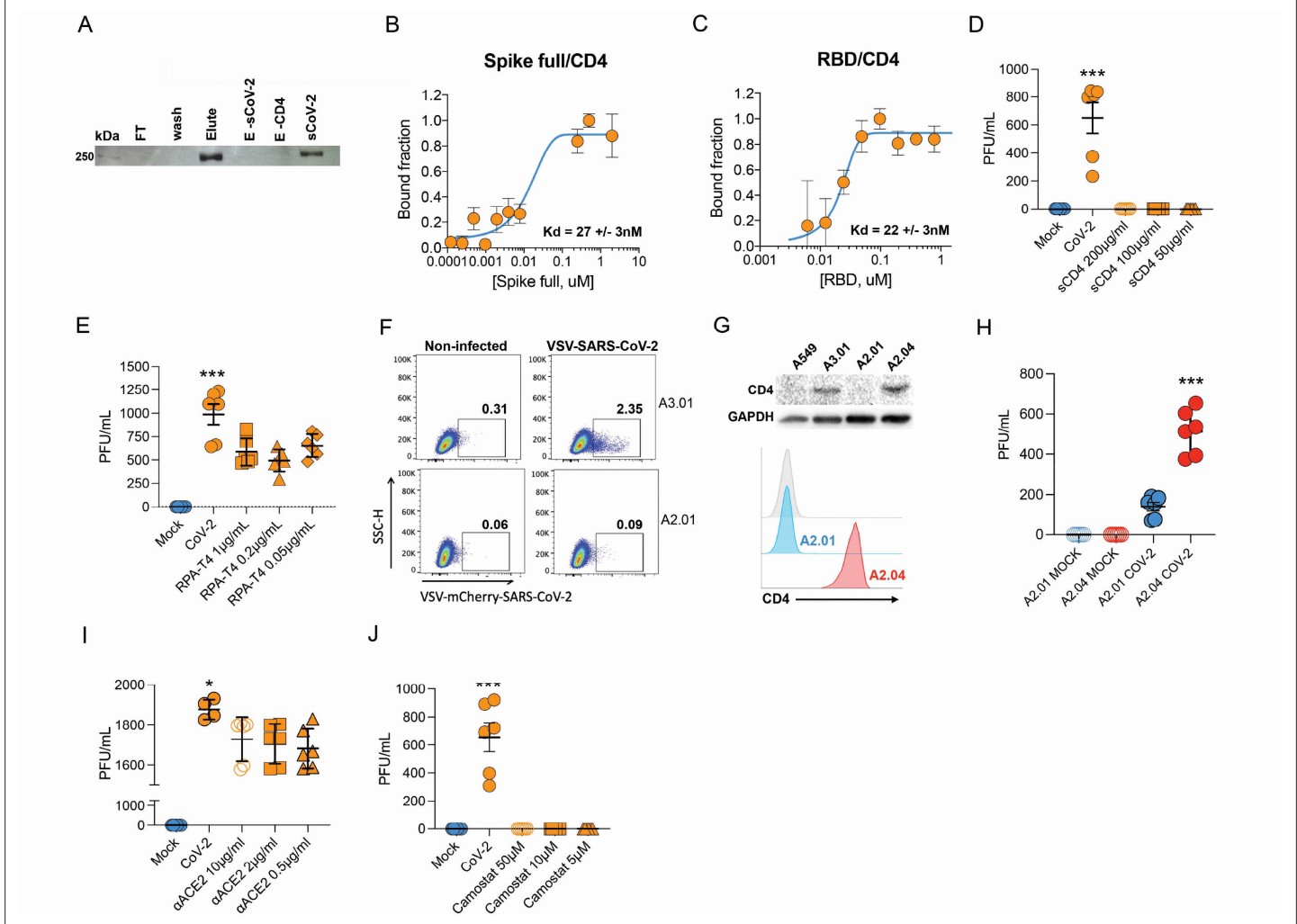

**Figure 3.** Infection of CD4+ T cells by SARS-CoV-2 is dependent on CD4 and ACE2 molecules. (**A**) Recombinant sCoV-2 (with twin-strep-tag) and CD4 were co-incubated and immunoprecipitated with anti-CD4. Complex formation was determined by Affinity blotting using streptavidin-HRP. (**B, C**) Fluorescence anisotropy curves of Spike full length and Spike receptor binding domain (RBD) binding to CD4 labelled protein, presenting dissociation constants (Kd) of 27 ± 3 nM and 22 ± 3 nM, respectively. The CD4 was labeled with FITC by incubation with fluorescein isothiocyanate probe, at molar ratio 3FITC:1protein, 4°C for 3 hr. The probe excess was removed by a desalting column (HiTrap 5 ml, GE) in a buffer containing 137 mM NaCl, 10 mM Na phosphate, 2.7 mM KCl, pH of 7.4. To evaluate binding affinities, serial dilutions of Spike (RBD or full length) were performed, from 4.5 μM to 100 nM, over 20 nM of labeled CD4. The measurements were taken using ClarioStar plate reader (BMG, using polarization filters of 520 nm for emission and of 495 nm for excitation) and data analysis were performed using OriginPro 8.6 software. The Kds were obtained from data fitted to binding curves through the Hill model. (**D**) Cells were exposed to mock control or SARS-CoV-2 (CoV-2) in the presence of vehicle or soluble CD4 (sCD4) in different concentrations (200, 100, or 50 μg/ml) for 1 hr under continuous agitation. Viral load was assessed by RT-qPCR 24 hr after infection. (**E**) Primary human CD4+ T cells were incubated with IgG control, or monoclonal anti-CD4 (RPA- T4) antibody 18 hr prior to exposure with mock control or CoV-2. Viral load was determined by RT-qPCR 24 hr after infection. (**F**) A2.01 and A3.01 lineages were cultivated with pseudotype virus (VSV-mCherry-CoV-2). Percentage of infected cells and flow cytometry analysis. (**G**) CD4 abundance by western blotting (upper panel) and flow cytometry (lower panel). (**H**) Viral load of CoV-2 in A2.01 and A2.04. (**I**) Peripheral blood CD4+ T cells were incubated with IgG control or anti-ACE2 (αACE2) polyclonal antibody 18 hr prior to exposure with mock control or CoV-2 for 1 hr. Viral load was determined 24 hr after infection. (**J**) CD4+ T cells were incubated with vehicle or camostat mesylate for 18 hr before the exposure with mock control or CoV-2 for 1 hr. Viral load was analyzed by RT-qPCR. Data represents mean ± SEM of at least two independent experiments performed in triplicate. *p<0.05, **p<0.01, ***p<0.0001.

The online version of this article includes the following source data and figure supplement(s) for figure 3:

**Source data 1.** Original co-immunoprecipitation blot.

**Source data 2.** Original ACE2 abundance by western blotting in T cell lines.

**Source data 3.** Original CD4 abundance by western blotting in T cell lines.

**Source data 4.** Original GAPDH abundance by western blotting in T cell lines.

*Figure 3 continued on next page*

*Figure 3 continued*

**Figure supplement 1.** Implication of ACE2 and CD4 expression profile for SARS-CoV-2 infection dynamics.

**Figure supplement 1—source data 1.** Western blotting of ACE2 in Caco-2, A2.01, A3.01, and CD4+ T lymphocytes.

**Figure supplement 1—source data 2.** Western blotting of CD4 in Caco-2, A2.01, A3.01, and CD4+ T lymphocytes.

**Figure supplement 1—source data 3.** Western blotting of B-actin in Caco-2, A2.01, A3.01, and CD4+ T lymphocytes.

The expression and release of IL-10 have been widely associated with chronic viral infections and determines viral persistence (*Brooks et al., 2006*). Noteworthy, increased serum levels of IL-10 are associated with COVID-19 severity (*Han et al., 2020*; *Zhao et al., 2020*). We found that IL-10 expression by CD4+ T cells was higher in BAL (*Figure 4—figure supplement 3*) and blood (*Figure 4C*) of severe COVID-19 patients. These changes were at least in part cell autonomous since purified CD4+ T cells exposed to SARS-CoV-2 also expressed higher levels of IL-10 (*Figure 4D*). Due to the immunomodulatory role of IL-10, we measured the expression of key pro- and anti-inflammatory cytokines involved in the immune response elicited by CD4+ T cells. CD4+ T cells from severe COVID-19 patients had decreased expression of IFNγ and IL-17A in relation to cells from patients with the moderate form of the disease or healthy donors (*Figure 4C*). These results show that SARS-CoV-2 induces IL-10 expression in CD4+ T cells, which is associated with suppression of genes that encode key pro-inflammatory cytokines, such as IFNγ and IL-17A, and correlates with disease severity.

The activation of the transcription factor CREB-1 via Ser$^{133}$ phosphorylation induces IL-10 expression (*Gee et al., 2006*). Consistent with IL-10 upregulation, CREB-1 phosphorylation at Ser$^{133}$ was increased in SARS-CoV-2-infected CD4+ T cells (*Figure 4E*). Thus, SARS-CoV-2 infection triggers a signaling cascade that culminates in the upregulation of IL-10 in CD4+ T cells. Altogether, our data demonstrate that SARS-CoV-2 infects CD4+ T cells, impairs cell function, leads to increased IL-10 expression, and compromises cell viability, which in turn dampens immunity against the virus and contributes to disease severity.

Impaired innate and adaptive immunity is a hallmark of COVID-19, particularly in patients that progress to the critical stages of the disease (*Hadjadj et al., 2020*; *Mathew et al., 2020*). Here, we propose that the alterations in immune responses associated with severe COVID-19 are at least partially triggered by infection of CD4+ T helper cells by SARS-CoV-2 and consequent dysregulation of immune function. T helper cells are infected by SARS-CoV-2 using a mechanism that involves binding of sCoV-2 to CD4 and entry mediated by the canonical ACE2/TMPRSS2 pathway. We propose a model where CD4 stabilizes SARS-CoV-2 on the cell membrane until the virus encounters ACE2 to enter the cell. Moreover, Cecon and colleagues showed that CD4 co- expression with ACE2 in HEK293 cells decreases the affinity and the maximal binding between the sCoV-2 RBD and ACE2 (*Cecon et al., 2022*), which is in line with our MD results and the binding assays (*Figure 3*), suggesting an intricate mechanism of modulation of SARS- CoV-2 infection in CD4+ T cells (*Figure 2C and D* and *Figure 2—figure supplement 1*). Taken together, these data indicate that in CD4-expressing cells direct binding of sCoV-2 to the CD4 molecule represents a potential mechanism to favor viral entry.

Once in CD4+ T cells, SARS-CoV-2 leads to protein expression changes consistent with alterations in pathways related to stress response, apoptosis, and cell cycle regulation, which, if sustained, culminate in cell dysfunction and may lead to cell death. SARS-CoV-2 also results in phosphorylation of CREB-1 transcription factor and induction of its target gene IL-10 in a cell-autonomous manner. IL-10 is a powerful anti-inflammatory cytokine and has been previously associated with viral persistence (*Brooks et al., 2006*). Serum levels of IL-10 increase during the early stages of the disease – when viral load reaches its peak – and may predict COVID-19 outcome (*Han et al., 2020*; *Zhao et al., 2020*). This increase occurs only in patients with the severe form of COVID-19 (*Zhao et al., 2020*). In contrast, we found IFNγ and IL-17A to be upregulated in CD4+ T cells of patients with moderate illness, indicating a protective role for these cytokines. However, in patients with severe illness, the expression of IFNγ and IL-17A in CD4+ T cells is dampened. IL-10 is a known suppressor of Th1 and Th17 responses and it is likely to contribute to the changes in IFNγ and IL-17A. These features will ultimately reflect in the quality of the immune response, which in combination with T cell death and consequent lymphopenia, may result in transient/acute immunodeficiency and impair adaptive immunity in severe COVID-19 patients (*Arunachalam et al., 2020*; *De Biasi et al., 2020*; *Thoms et al., 2020*). Lymphocytopenia is a characteristic feature of severe COVID-19 (*Zheng et al., 2020*). Here, we demonstrated that infected

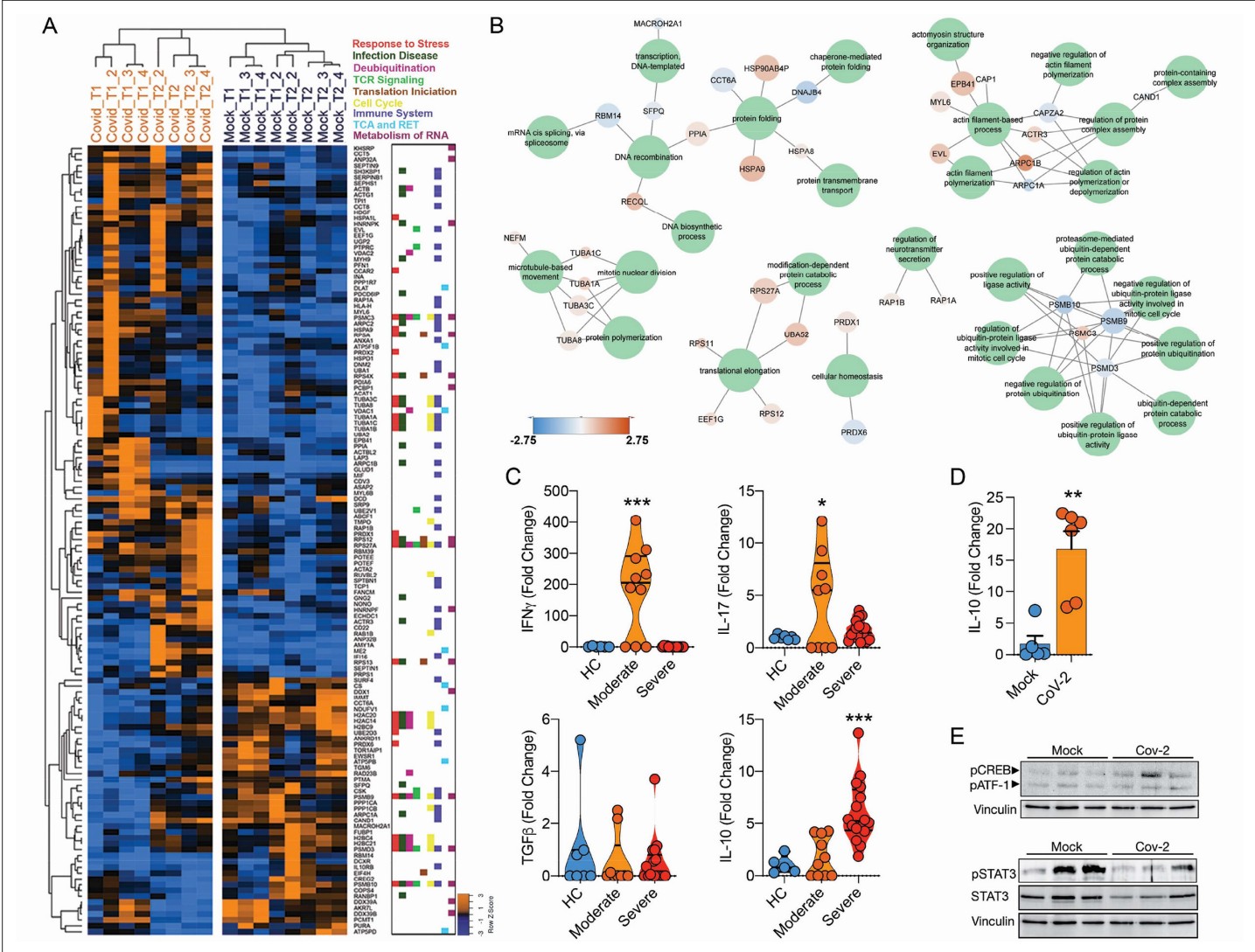

**Figure 4.** Infection of CD4[+] T cells by SARS-CoV-2 alters cell function and triggers IL-10 production. (**A**) Heatmap of differentially expressed proteins and their associated biological processes. (**B**) Network of differentially expressed proteins and their associated biochemical pathways. (**C**) Relative expression of *IFNγ, IL-17A, TGFβ,* and *IL-10* genes in peripheral blood CD3[+] CD4[+] cells from COVID-19 patients (moderate or severe) and healthy control (HC). Data represents mean ± SEM. Each dot representes a patient sample. (**D**) Relative expression of *IL-10* gene in primary CD4[+] T cells infected with SARS-CoV-2. (**E**) Representative immunoblotting of phosphorylated CREB-1Ser133 (pCREB) and total or phosphorylated STAT3 in peripheral blood CD3[+] CD4[+] exposed to mock control or SARS-CoV-2. Phosphorylated ATF-1 is detected by the same antibody used to detect pCREB. ***p<0.001, **p<0.01, *p<0.05.

The online version of this article includes the following source data and figure supplement(s) for figure 4:

**Source data 1.** Original immunoblotting of phosphorylated CREB-1Ser133 (pCREB) and phoshorylated ATF-1.

**Source data 2.** Original immunoblotting of phosphorylated STAT3.

**Source data 3.** Original immunoblotting of total STAT3.

**Source data 4.** Original immunoblotting of Vinculin.

**Figure supplement 1.** Analysis of differentially expressed proteins and their impact on biological processes.

**Figure supplement 2.** Functional protein network and enrichment analysis in infected CD4[+] T cells.

**Figure supplement 3.** Integrated analysis of BAL CD4[+] T cells shows immune dysregulation in COVID-19 patients.

CD4$^+$ T cells undergo cell death, which may contribute to the development of lymphocytopenia. In addition, cytokines such as IL-1β released by infected monocytes can promote T cell dysfunction (*Codo et al., 2020*). Therefore, the decrease in lymphocyte count may be attributed to both infected T cell death and dysfunctional T cells resulting from inflammatory cytokine storm.

How long these alterations in T cell function persist in vivo and whether they have long-lasting impacts on adaptive immunity remains to be determined. Hence, avoiding T cell infection by blocking sCoV-2-CD4 interaction and boosting T cell resistance against SARS-CoV-2 might represent complementary therapeutic approaches to preserve immune response integrity and prevent patients from progressing to the severe stages of COVID-19.

## Methods

### Subjects
The samples from patients diagnosed with COVID-19 were obtained at the Clinical Hospital of the University of Campinas (SP-Brazil). Both COVID-19 patients and healthy subjects included in this work signed a term of consent approved by the University of Campinas Committee for Ethical Research (CAAE: 32078620.4.0000.5404; CAEE: 30227920.9.0000.5404). Human blood samples from severe COVID-19 patients analyzed in this study were obtained from individuals admitted at the Clinics Hospital, University of Campinas, and included in a clinical trial (UTN: U1111-1250-1843). In vitro experiments were performed with buffy coats from healthy blood donors provided by The Hematology and Hemotherapy Center of the University of Campinas (CAAE: 31622420.0.0000.5404).

### Diagnosis
Nasopharyngeal swabs or bronchoalveolar lavage fluids (BALFs) were tested for SARS-CoV-2 by real-time qPCR. The tests were performed in the Laboratory of High-Performance Molecular Diagnostic or Laboratory of Clinical Pathology, all located at the University of Campinas. The samples were aliquoted and extracted manually using the MagMAX Viral/Pathogen Nucleic Acid Isolation Kit (Applied Biosystems). Some RNAs were automatically isolated by MagMAX Express-96 Magnetic Particle Processor (Applied Biosystems) following the manufacturer's protocol. Real-time qPCRs were performed in duplicates using TaqMan Fast Virus 1-Step Master Mix (Applied Biosystems) for the detection of SarbecoV E-gene with specific primers (10 µM) and probe (5 µM). Primer sequences were: forward-ACAGGTACGTTAATAGTTAATAGCGT; reverse-ATATTGCAGCAGTACGCACACA, Probe:-6FAM-ACACTAGCCATCCTTACTGCGCTTCG- QSY. Patients were considered positive for COVID-19 when Ct < 38. qPCRs were performed using the Applied Biosystems 7500 Fast Real-time PCR system.

### Blood sample collection and lymphocyte separation
Each COVID-19 patient had heparin and plain blood tubes collected. Whole blood, serum, and plasma samples were separated. PBMCs from COVID-19 patients and buffy coats were obtained through the Histopaque-1077 density gradient (Sigma-Aldrich). Samples were diluted in Hanks (1:1) and gently poured into 15 or 50 ml conical tubes containing 3 or 10 ml of Histopaque, respectively. Then, the samples were centrifuged at 1400 rpm for 30 min at 4°C without brake. The PBMC layer was collected into a new tube, lymphocytes were sorted (see below), and incubated overnight at 37°C with 5% CO$_2$ atmosphere with RPMI 1640 (Gibco) containing 10% fetal bovine serum (FBS) and 1% Penicillin-Streptomycin (P/S).

### A2.01, A2.04, and A3.01 cell lines
The A3.01 (ARP-166) and A2.01 (ARP-2059) cells were acquired from the NIH AIDS Reagent Program (MD) and maintained in RPMI 1640 (Gibco) containing 10% FBS and 1% P/S at 37°C with 5% CO$_2$ atmosphere. Mycoplasma contamination was not detected. A3.01 is a lymphoblastic leukemia cell line that expresses high levels of CD4, while A2.01 does not show detectable levels of CD4. The A2.04 was produced for this article. In brief, the coding sequence for CD4 was amplified from pCMV.CD4 and cloned into the BglII/SalI cloning site of pMSCV-IRES-GFP. The retrovirus carrying pMSCV-CD4-IRES-GFP vector was produced in Peak cells by using pVSV-G and pCL-Eco packing vectors as previously described (*Tavares et al., 2016*; *Tavares et al., 2020*). The A2.01 cells (*Folks et al., 1986*) were transduced with this retrovirus and sorted for GFP positive in the BD FACSCanto II flow cytometer

in the Ribeirão Preto Center for Cell-Based Therapy/Hemotherapy, and the confirmation for CD4 expression was done by western blot and flow cytometry.

## Virus and infection

The ancient SARS-CoV-2 viral lineage [CoV-2(B)] was isolated from the second confirmed case of COVID-19 in Brazil and kindly donated by Prof. Dr. Edson Luiz Durigon (University of São Paulo, Brazil; SARS.CoV.2/SP02.2020, GenBank accession number MT126808). The SARS-CoV-2 gamma variant [CoV-2(P.1)] was isolated from residual nasopharyngeal lavage specimens of a patient from Manaus City, Brazil, who tested positive for SARS-CoV-2 (GISAID accession ID EPI_ISL_1708318). SARS-CoV-2 virus stocks were propagated in Vero cell line CCL-81 (STR profiling authentication method, mycoplasma contamination not detected, origin: ATCC) and supernatant was harvested at 2 dpi. The viral titers were determined by plaque assays on Vero cells. Vero cells were propagated in Dulbecco's modified Eagle's medium (DMEM) supplemented with 10% FBS, 1% P/S, and maintained at 37°C in a 5% $CO_2$ atmosphere. Cells were incubated with the CoV-2(B) or CoV-2(P.1) lineage (MOI = 1 unless indicated otherwise) for 1 hr at room temperature. After viral adsorption, cells were washed three times with PBS and incubated with a culture medium under standard culture conditions (37°C and 5% $CO_2$ atmosphere). Most of the in vitro experiments were done with 1 million sorted CD3$^+$CD4$^+$ live cells plated over a well of a 48-well plate. For proteomic and immunoblotting experiments, 4 million live CD3$^+$CD4$^+$ cells were used. All experiments with SARS-CoV-2 were done in a biosafety level 3 laboratory (BSL-3) located in the Institute of Biology from the University Campinas.

## Flow cytometry

CD4$^+$ T cells were sorted with FACSJazz or FACS Melody (BD Biosciences) as CD4$^+$CD3$^+$ and, in some experiments, CD8$^-$CD3$^+$. CD8$^+$ T cells were sorted as CD8$^+$CD3$^+$ or CD4$^+$CD3$^+$ (see the list of antibodies in *Supplementary file 3*). We performed flow cytometry with a lineage marker, Lin CD3, CD14, CD16, CD19, CD20 and CD56 to avoid contaminations. Sorted cells were incubated overnight in RPMI 1640 media containing 10% FBS and 1% P/S at 37°C with 5% $CO_2$ atmosphere for infection. All flow cytometry analyses were performed in FACS Symphony A5 (BD Biosciences). All antibodies used in the present study were sodium azide free.

## Biotinylated DNA probe synthesis

Probe synthesis is based on two PCR steps both using the same pair of primers (primer sense: AACA CGCAAATTAATGCCTGTCTG; primer antisense: GTAACAGCATCAGGTGAAGAAACA) for the RdRp gene. The first amplifying PCR mix contains 1× buffer, 1.5 mM MgCl$_2$, 0.2 mM dNTP, 0.1 μM of each primer, 1 μl of cDNA (1:10), and 1.5 U Taq polymerase (GoTaq DNA Polymerase, Promega) in a final volume of 50 μl. The reaction was subjected to 35 cycles of PCR including denaturation at 95°C for 30 s, annealing at 55°C for 30 s, and extension at 72°C for 30 s, following incubation at 72°C for 5 min. PCR samples were analyzed by electrophoresis on a 1% agarose gel in order to verify a final amplicon of 300 pb. The second PCR contained slight modifications in relation to the first, that is, we used 0.2 mM of d (C, G, A) TP and 0.2 μM of 16-dUTP instead of the regular dNTP mix, and used 1 μl of the previous PCR as a template (*Day et al., 1990*).

## Plaque assays

Vero cells were grown in 24-well plates up to 80% confluency. Supernatant samples from infected lymphocytes were added and incubated at 37°C with 5% $CO_2$ for an hour for virus adsorption. Samples were replaced with a semi-solid overlay medium (1% w/v carboxy methyl cellulose in complete DMEM) and incubated at 37°C with 5% of $CO_2$ for 3–4 d. Plates were fixed in 10% w/v paraformaldehyde (PFA), stained in 1% w/v methylene blue, and results were expressed as viral plaque-forming units per milliliter of sample (PFU/ml).

## Co-immunoprecipitation

Co-immunoprecipitation was performed using Pierce protein A/G magnetics beads (Cat# 88803, Thermo Fisher Scientific) immobilized with anti-CD4 antibody (Cat# IM0566, Rhea Biotech). Briefly, 100 μg of magnetic bead was washed in PBS containing 0.05% Tween-20 and then incubated with 10 μg anti-CD4 antibody for 2 hr at 4°C with mixing. For complex formation, we co-incubated 60

pmoles of recombinant CD4 (Cat# 10400-H08H, Sino Biological) and 60 pmoles of Spike S1 Twin-Strep-Tag recombinant protein expressed and generously provided by Dr. Leda Castilho from the Federal University of Rio de Janeiro (*Alvim et al., 2020*). Incubation was performed in 300 µl of PBS and kept overnight at 4°C with gentle agitation. After incubation, the recombinant protein solution was transferred to a 1.5 ml tube with the washed immobilized beads and incubated for 1 hr at room temperature. The complex was precipitated and proteins that did not bind to the beads were subsequently removed during the washes. The protein was eluted from the bead using Laemmli buffer and heat (95° for 5 min). The sample was analyzed by affinity blot using a streptavidin-HRP detection (Cat# 405210, BioLegend). Full-range Rainbow (Cat# RPN800E) was used as the molecular weight marker.

## Spike target prediction

The P-HIPSTER database (http://phipster.org; last accessed: June 1, 2020) was used to find potential SARS-CoV-2 spike-human interactions. P-HIPSTER discoveries are predicted using protein similarities of known interactions (*Lasso et al., 2019*). Only interactions with a final likelihood ratio (LR) score ≥100 were used in the analysis. According to the authors, this threshold has a validation rate of 76%. To evaluate whether these proteins were found on the cellular membrane, four databases were used. The Human Protein Atlas (https://www.proteinatlas.org; last accessed: June 1, 2020) contains tissue and anatomical protein annotations; 1917 proteins classified as 'Cell Membrane' were retrieved. Panther GO is a gene set enrichment analysis tool (Mi H, PANTHER version 14: More genomes, a new PANTHER GO-slim and improvements in enrichment analysis tools); 1504 genes annotated as 'integral components of plasma membrane' (GO:0005887) were retrieved. EnrichR (https://amp.pharm.mssm.edu/Enrichr/ ;last accessed: June 1, 2020) is a web-based app that integrates various enrichment analysis tools (*Kuleshov et al., 2016*), which library was used to retrieve JENSEN compartment annotations (https://compartments.jensenlab.org/; last accessed: June 1, 2020) for 'external side of plasma membrane' (212), 'plasma membrane' (1148), and 'cell surface' (715). To find common elements between the datasets and generate the Venn diagram, jvenn (http://jvenn.toulouse.inra.fr/app/; last accessed: June 1, 2020) was used (*Bardou et al., 2014*).

## Gene set enrichment analysis

EnrichR was used for enrichment analysis for SARS-CoV-2 spike protein predicted interactions. GO Molecular Function data was downloaded, and enrichments with an adjusted p-value<0.05 were retrieved.

## Global expression analysis of ACE2, TMPRSS2, and CD4

To evaluate which tissues and cells express ACE2, TMPRSS2, and CD4, expression data (both protein and consensus mRNA expression, NX) from The Human Protein Atlas (https://www.proteinatlas.org; last accessed: July 7, 2020) (*Thul and Lindskog, 2018*) and microarray expression from BioGPS (http://biogps.org/#goto=welcome; last accessed: Jul 7, 2020) (*Wu et al., 2016*) were downloaded and plotted. In The Human Protein Atlas data, only normalized expression level ≥1 and protein detection ≥ 'low' of at least one of the three genes was plotted. In the BioGPS data, this threshold was ≥10 a.u. (arbitrary units).

## sCoV-2-ACE2-interacting peptide (βACE2) design and synthesis

A peptide representing the interacting region of SARS-CoV-2 Spike protein with ACE2 and present at the receptor binding domain (RBD receptor binding doming) was designed based on the tridimensional structure described elsewhere (*Shang et al., 2020*). Based on the structure, we selected an unstructured and continuous segment of 16 amino acid residues (THR487-GLY502), which contained most of the interaction points with ACE2. We added to the segment a substitution of CYS488 to SER, which intended to avoid random disulfide bond formation and structural alterations, keeping the same hydrophilicity in the interacting region. An amidated CYS residue was added at the C-terminal of the peptide to allow simple and specific conjugation with accessory detection molecules. The complete 17 amino acid residues (487NCYFPLQSYGFQPTNG502C) were synthesized using standard FMOC solid-phase peptide synthesis chemistry as previously described (*Fields and Noble, 1990*) at a 100 µM scale using RINK-amide resin 0.7 mmol/g (Advanced ChemTech, Cat# SA5130). All FMOC-aas (Advanced ChemTech) were used with 2.5 excess. FMOC-aa coupling reaction assisted by six cycles of

2 min in a home-made microwave device. At the end of coupling reactions, the peptide was cleaved from the resin using a solution of 88% trifluoroacetic acid solution, 4% water, 4% triisopropylsilane, 2% anisole + 30 mg dithiothreitol for 2 hr. Cleaved peptide was precipitated with ethylbutylether and then purified on C18 seppack solid phase extraction cartridges (10 mg, Sigma-Aldrich) using a water:acetonitrile solvent system.

## Proteomic analysis

Infected and mock cells were resuspended in RIPA buffer (150 mM NaCl, 1 mM EDTA, 100 mM Tris-HCl, 1% Triton X-100) with freshly added protease and phosphatase inhibitors (Protease Inhibitor Cocktail, Sigma). After three cycles of 30 s of ultra-sonication for cell lysis, the protein amount was quantified by the Pierce BCA protein assay kit (Thermo Scientific).

Aiming to obtain a higher yield, we performed the FASP protocol for subsequent analyses (*Distler et al., 2016*). The FASP protocol is a method that allows to concentrate proteins and clean up the samples through washing steps in a microcolumn tip with a 10 kDa MW cut off and perform the tryptic digestion in this column. Also, 10 µg of protein were used to carry out the FASP protocol, where the samples were reduced, alkylated, and later digested using trypsin. An amount of 100 fmol/µl of digestion products of Enolase from *Saccharomyces cerevisiae* was added to each sample as internal standard, then separation of tryptic peptides was performed on an ACQUITY MClass System (Waters Corporation). Then, 1 µg of each digested samples was loaded onto a Symmetry C18 5 µm, 180 µm × 20 mm precolumn (Waters Corporation) and subsequently separated by a 120 min reversed phase gradient at 300 nl/min (linear gradient, 2–85% $CH_3CN$ over 90 min) using a HSS T3 C18 1.8 µm, 75 µm × 150 mm nanoscale LC column (Waters Corporation) maintained at 40°C. After peptide separation, the ionized peptides were acquired by a Synapt G2-Si (Waters Corporation). Differential protein expression was evaluated with data-independent acquisition (DIA) of shotgun proteomics analysis by Expression configuration mode using the Ion Mobility cell (HDMSe). All spectra have been acquired in Ion Mobility Mode by applying a wave velocity for the ion separation of 800 m/s and a transfer wave velocity of 175 m/s. The mass spectrometer operated in 'Expression Mode,' switching between low (4 eV) and high (25–60 eV) collision energies on the gas cell, using a scan time of 0.5 s per function over 50–2000 m/z. The processing through low and elevated energy, added to the data of the reference lock mass ([Glu1]-Fibrinopeptide B Standard, Waters Corporation) provides a time-aligned inventory of accurate mass-retention time components for both low and elevated-energy (exact mass retention time [EMRT]).

Each sample was analyzed in four technical replicates. Continuum LC-MS data from three replicate experiments for each sample was processed for qualitative and quantitative analysis using the software Progenesis QC for Proteomics (PLGS, Waters Corporation). The qualitative identification of proteins was obtained by searching in the *Homo sapiens* database (UniProtKB/Swiss-Prot Protein reviewed 2020). The expression analysis was performed considering technical replicates available for each experimental condition following the hypothesis that each group is an independent variable. Protein identifications were based on the detection of more than two fragment ions per peptide, and more than two peptides measured per protein. The list of normalized proteins was screened according to the following criteria: protein identified in at least 70% of the runs from the same sample and only modulated proteins with a $p < 0.05$ were considered significant. Raw data are available in ProteomeXchange database under accession number PXD020967.

## In situ hybridization

Sorted CD4[+] T cells were incubated for 24 hr with the SARS-CoV-2 virus (MOI of 1) or mock. After 24 hr, the cells were washed three times with PBS and fixed with 4% PFA for 15 min, at room temperature, followed by two washes with PBS (pH 7.4) prepared with diethylpyrocarbonate (DEPC)-treated water. Further, the cells were seeded on silanized glass slides and kept on a prewarmed surface until liquid is fully evaporated. The slides were washed with PBS and pretreated with 2% $H_2O_2$ in methanol for 30 min to bleach auto- and avoid nonspecific fluorescence. They were then washed twice with PBS 0.05% Triton X-100 (PBST), treated with Proteinase K (10 µg/ml) for 2 min, followed by PBST-Glycin (2 mg/ml) for 10 min, and a second fixation step with PFA 4% for 10 min, followed by three more washes with PBST at room temperature. Each sample was pre-hybridized with hybridization solution (Hyb) (50% formamide, 10% dextran sulfate, 2× saline-sodium citrate buffer [SSC] pH 7.0,

and 100 µg salmon DNA), without the probe, for at least 2 hr at 37°C in a humid box. Samples were then incubated with Hyb containing the biotinylated probe (100 µl of the probe in 500 µl of Hyb), pre-denatured (85°C for 10 min), in a humid box overnight at 37°C. The slides were then subjected to sequential washes with 50% Hyb in 2× SSC and 25% Hyb in 2× SSC, for 20 min at 37°C each, and two washes for 10 min each with 2× SSC, 0.2× SSC, and PBS. All samples were incubated with fluorescent streptavidin (Streptavidin, DyLight 594 Conjugated, Thermo Scientific, #21842, 1:300) for 2 hr at room temperature in the humid box protected from light. The slides were washed with PBST, incubated with DAPI (Santa Cruz Biotechnology, SC-3598), diluted 1:1000 in PBST for 5 min at room temperature, protected from light, and mounted in an aqueous mounting solution for confocal imaging.

## Detection of SARS-CoV-2 antisense strand

cDNA was generated using SuperScript III First-Strand Synthesis SuperMix (Thermo Fisher) according to the manufacturer's instructions. Then, 60 ng of RNA were incubated at 65°C for 5 min with 1× annealing buffer and 250 nM of a primer complementary to the anti-sense CoV-2 strand containing a nonviral tag at the 5' end. The sample was chilled at 4°C for 1 min and incubated at 55°C for 50 min with SuperScript III/RNaseOUT and 1× master mix for reverse transcription (RT). The sample was heated to 85°C for enzyme inactivation. The antisense strand was amplified in a PCR using 100 nM of primers complementary to the nonviral tag and to the antisense CoV-2 strand. As a control for false-priming (*Bessaud et al., 2008*), we performed the RT without the tagged reverse primer and a PCR using both the RT and the PCR primers at 100 nM. The sense strand was also amplified as a control with the same method using nontagged primers for the RT and PCR. The samples were loaded in a 4% agarose gel in Tris/borate/EDTA buffer containing 1× SybrSafe (Thermo Fisher).

## Immunoblotting

Cells were infected for 3 hr with the SARS-CoV-2 virus (MOI of 1) or mock, washed with PBS, and the protein was extracted using lysis buffer (Tris-HCl 100 mM pH 7.5; EDTA 1 mM; NaCl 150 mM; 1% Triton X-100; 1× protease and phosphatase inhibitors). Protein quantification was performed using Pierce BCA protein assay kit (Thermo Fisher Scientific). Then, 5 µg protein of each sample was diluted with Laemmli buffer and heated at 95°C during 5 min. Samples were subjected to 10% SDS-PAGE and transferred to PVDF membrane (Cat# 1620177, Bio-Rad). Membranes were blocked with Starting-Block (37539, Thermo Fisher Scientific) for 30 min, and incubated overnight with anti-phospho-CREB (Cat# 9198, Cell Signaling, dilution 1:1000), anti-phospho-STAT3 (Cat# 8059, Santa Cruz, dilution 1:250), anti-STAT3 (Cat# 4904, Cell Signaling, dilution 1:1000), anti-ACE-2 (Cat# ab15348, Abcam, dilution 1:2000), or anti-CD4 (Cat# sc-7219, Santa Cruz, dilution 1:2000). Membranes were washed and incubated during 1 hr with rabbit IgG horseradish peroxidase-conjugated secondary antibodies (Cat# RPN4301, GE Healthcare, diluted 1:10,000 in block buffer). Membranes were then washed and incubated with Immobilon Western Chemiluminescent HRP Substrate (Cat# WBKLS0500, Millipore), and the images were acquired in ChemiDoc Gel Imaging System (Bio-Rad). Anti-Vinculin (Cat#, ab91459, Abcam, dilution 1:1000), anti-actin-beta (Cat#, sc-47778, Santa Cruz, dilution 1:4000), and anti-GAPDH (Cat# G9545, Sigma-Aldrich, dilution 1:5000) antibodies were used as loading controls. Quantification was performed using ImageJ.

## Immunofluorescence

Cells were prepared onto silanized glass slides as previously described for in situ hybridization (*Alamri et al., 2017*), followed by a washing step using PBST 0.1 M pH 7.4. To avoid autofluorescence, the cells were treated with 2% $H_2O_2$ in methanol for 30 min, washed with PBST, and treated with 0.1 M glycine in PBST for 10 min at room temperature. The samples were then washed and treated with 1% bovine serum albumin (BSA) solution in PBST for 30 min (according to the manufacturer's suggestion), to block nonspecific epitopes. Cells were incubated with SARS-CoV-2 Spike S1 Antibody (HC2001) (GenScript, A02038) diluted 1:100 in 1% BSA solution in PBST, and incubated overnight at 4°C in a humid box. The slides were then washed and incubated with anti-human IgG Alexa 488 (Thermo Fisher, A11013) diluted 1:500 in 1% BSA solution in PBST for 2 hr at room temperature in a humid box, protected from light. The samples were washed again, incubated with DAPI (Santa Cruz Biotechnology, SC-3598) diluted 1:1000 in 1% BSA solution in PBST for 5 min at room temperature protected from light, and mounted in an aqueous mounting solution for confocal imaging. Microscopy images

were acquired with a Zeiss LSM880 with Airyscan on an Axio Observer 7 inverted microscope (Carl Zeiss AG, Germany) with a C Plan-Apochromat ×63/1.4 Oil DIC objective, 4× optical zoom. Prior to image analysis, raw .czi files were automatically processed into deconvoluted Airyscan images using the Zen Black 2.3 software. For DAPI, conventional confocal images were acquired using a 405 nm laser line for excitation and a pinhole set to 1 AU.

## Transmission electron microscopy

T lymphocyte cell cultures were pelleted by centrifugation at 500 × *g* for 10 min. Cell culture supernatant was removed and cells were resuspended in 100 µl of the fixative solution, which consisted of 1 M sodium cacodylate aqueous solution supplemented with 2.5 % v/v glutaraldehyde and 3 mM calcium chloride at pH 7.2, and kept overnight (16 hr) at 4°C. Cells were pelleted (1500 rpm for 2 min) and washed in cacodylate buffer with calcium chloride for a total of five times. Post-fixation was performed with 1% reduced osmium tetroxide plus 0.8% potassium ferrocyanide in a cacodylate buffer with calcium chloride for 2 hr at 4°C covered from light. Pellet was recovered by centrifugation at 1500 rpm at 4°C and washed three times in ddH$_2$O for 2 min. Cells were resuspended in 4% low melting agarose (approximately 50°C) and centrifuged at 1500 rpm for 10 min at 30°C and kept 20 min on ice for solidification. Agarose pellets were trimmed in small blocks and stained for contrast enhancement with 2% aqueous uranyl acetate overnight at 4°C. Later, samples were washed in ddH$_2$O and dehydrated in increasing concentrations of ethanol (20, 50, 70, 80, 90, 100% twice), 1:1 ethanol:acetone, and acetone (twice), 20 min each. Samples were embedded in Embed-812 (Electron Microscopy Science, USA, #14120) following manufacturer's recommendations. Ultrathin sections (70 nm) were mounted in 200 mesh copper grids and stained with 2% aqueous uranyl acetate and Reynold's lead citrate. Finished grids were imaged in a JEOL JEM-1400 Transmission Electron Microscope (120 kV accelerating voltage) and/or Helios Nanolab Dualbeam 660 Scanning Transmission Electron Microscope (30 kV accelerating voltage) at the Electron Microscopy Laboratory, Brazilian Nanotechnology National Laboratory, LNNano, CNPEM.

## RNA extraction cDNA synthesis

Total RNA was extracted from samples using TRI Reagent (Sigma) (200 µl per pellet of cells) according to the manufacturer's instructions and quantified using NanoDrop 200 Spectrophotometer (Thermo Scientific). cDNA was synthesized using 500 ng of total RNA and the GoScript Reverse Transcriptase Kit (Promega) or the High-Capacity cDNA Reverse Transcription Kit (Applied Biosystems) following the manufacturer's instructions. The cDNA final concentration was 25 ng/µl.

## qPCR

Real-time PCR (qPCR) was performed using QuantiNova SYBR Green PCR Kit (QIAGEN) with specific primers (*HPRT* – forward: GACCAGTCAACAGGGGACAT, reverse: AACACTTCGTGGGGTCCTTTTC; *IL10* – forward: GCCTAACATGCTTCGAGATC; reverse: CTCATGGCTTTGTAGATGCC; *TGFB1* – forward: AAGTTGGCATGGTAGCCCTT, reverse: GCCCTGGATACCAACTATTGC; *IFNG* – forward: TTTAATGCAGGTCATTCAGATGTA, reverse: CACTTGGATGAGTTCATGTATTGC; and *IL17A* – forward: TCCCACGAAATCCAGGATGC, reverse: TGTTCAGGTTGACCATCACAGT). Expression levels of each gene were normalized to a housekeeping gene (i.e., hypoxanthine phosphoribosyltransferase 1 [*HPRT1*]). qPCR reaction was performed using 15 ng of cDNA, 300 nM of specific primers, and QuantiNova SYBR Green (QIAGEN). Amplification occurred upon cycling conditions: 95°C for 3 min and 45 cycles of 95°C for 15 s, 60°C for 20 s, and 72°C for 30 s. Amplification plots, melting curves, and Ct values were obtained using CFX Manager (Bio-Rad). Gene expression fold change was calculated with the ΔΔCt method (*Supplementary file 4*). Viral load was determined using specific SARS-CoV-2 N1 primers, as described previously (*Won et al., 2020*) and using qPCRBIO Probe 1-Step Go Lo-Rox (PCR Biosystem) with specific SarbecoV E-gene primers and probe. For the preparation of a standard curve, serial dilutions of SARS-CoV-2 were used. qPCR was performed using the Bio-Rad CFX384 Touch Real-Time PCR Detection System and Thermo Fisher QuantStudio 3. Relative expression of the viral envelope (E), nucleocapsid (N), and RNA-dependent RNA polymerase (RdRp) genes from in vitro CD3$^+$CD4$^+$ cells were detected using GeneFinder COVID-19 Plus Real*Amp*Kit (OSANG Healthcare Co). Expression level of each gene was normalized to internal control of the kit and relative gene

expression was calculated as $2^{-\Delta CT}$. For this kit, the qPCR was performed using the Applied Biosystems 7500 Fast Real-time PCR system.

## Single-cell mRNA sequencing (scRNAseq)

We reanalyzed the scRNAseq data from six samples of BALF-containing immune cells of severe COVID-19 patients (C145, C146, C148, C149, and C152) by downloading the respective single-cell processed data matrices from GEO under the accession number GSE145926. The number of viral transcripts mapped to SARS-CoV-2 was previously integrated by their authors as an additional feature into data matrices called 'NcoV' used to quantify the viral load of SARS-CoV-2. The data matrices were then imported to R version 3.6.3 and analyzed using Seurat v3.1 (*Stuart et al., 2019*).

A quality control filtering was applied to remove low-quality cells considering a gene number between 200 and 6000, unique molecular identifier (UMI) count > 1000, and mitochondrial gene percentage <0.1. After the filtering step, a total of 37,197 cells and 25,722 features (including SARS-CoV-2 viral load) were left for downstream analysis. The filtered gene-barcode matrix of all samples was integrated with FindIntegrationAnchors and IntegrateData functions considering the first 50 dimensions of canonical correlation analysis (CCA). Then, the filtered data matrix was normalized using the 'LogNormalize' method in Seurat with a scale factor of 10,000. The top 3000 variable genes were then identified using the 'vst' method in Seurat using the FindVariableFeatures function. Variables 'nCount_RNA' and 'percent.mito' were regressed out in the scaling step, and principal component analysis was performed using the top 3000 variable genes using 100 dimensions. Then, Uniform Manifold Approximation and Projection and t-Distributed Stochastic Neighbor Embedding were performed on the top 40 principal components for visualizing the cells. Additionally, a clustering analysis was performed on the first 40 principal components using a resolution of 0.8. Then, differential gene expression analysis was performed with the Wilcoxon rank-sum test using the FindAllMarkers function to obtain a list of representative gene markers for each cluster of cells. Representative gene markers were considered based on adjusted p-value<0.05 and average log fold change > 0.25 among cells. T cell cluster annotation was performed using an evidence-based score approach with scCATCH (*Shao et al., 2020*) based on differential gene markers list. Finally, CD4$^+$ T cell subpopulations were annotated considering a high differential expression of CD4, CCR7, and IL32 gene markers (*Figure 1—figure supplement 5*). Based on the methodology applied above, a total of 1846 single cells were identified as CD4$^+$ T cells. T cells were considered infected assuming a threshold with at least 10% of viral RNA expression. The applied threshold resulted in 39 single cells infected with SARS-CoV-2.

## Docking

The RosettaDock4.0 protein docking protocol (*Leman et al., 2020*; *Marze et al., 2018*) was applied to model the interaction between CD4 NTD (*Wu et al., 1997*) (PDB ID 1wio, residues 1–178) and sCov2 RBD (*Wang et al., 2020*) (PDB ID 6lzg, residues 333–527) as described previously. Briefly, the first step consisted of generating an ensemble of 100 conformers for each target. The two-stage automated Rosetta docking protocol simulates the physical encounter of the proteins and maximizes interactions, which may lead to binding. In the first stage (global docking), a rigid-body translation-rotation of one of the protein samples' possible interaction modes using a low-resolution, centroid-based representation of the side chains. RosettaDock4.0 also mimics binding-induced conformational adaptations by assessing a pre-generated list of conformers. This is an important feature for cases where backbone perturbation is inherent to the interface of interaction. Additionally, a score term named Motif Dock Score ranks low-resolution models according to their chance to generate promising high-resolution models, reducing the number of candidates to proceed to the next stage. In the second stage (local docking), the centroid mode is converted into full-atom representation, where small random rigid-body perturbations take place, followed by side-chain relaxation that aims to optimize local interactions. A first set of models consisted of generating 10,000 low-resolution candidates using global docking with 3 Å translation and 8° rotation perturbation parameters. The 2000 highest-ranked models according to Motif Dock Score proceeded to the local docking stage using default parameters (0.1 Å translation and 3° rotation perturbation parameters). Twenty-five high-resolution candidates were generated for each low-resolution input model totaling 50,000 models. A second set of models was generated by first randomizing CD4 NTD orientation previously to docking run. For this strategy, 40,000 low-resolution and 300,000 high-resolution models were generated similarly to the

description above. As an alternative docking approach, monomeric chains were also docked using the HADDOCK 2.4 server (*van Zundert et al., 2016*), which returned 40 model candidates spread into 13 clusters. A total of 800 additional models were generated using these models as input to the high-resolution docking stage at RosettaDock4.0. Next, best-ranked 2000 models according to interface score (I_sc) were fully relaxed with Relax application (beta_genpot score function, Rosetta Package) and evaluated with InterfaceAnalyzer application (Rosetta package) to extract interface metrics. A total of 500 top-ranked models according to interface binding energy (dG_separated) were evaluated for pairwise structural similarity with LovoAlign (*Martínez et al., 2007*) and were clustered at 2.0 Å radius. Best-ranked models of each of the first 50 clusters were selected for MD simulation. This set of models is available at https://github.com/ajrferrari/CD4-RBD-interaction-models (copy archived at *Ferrari, 2023*).

## Molecular dynamics simulations

MD simulations have been used to discriminate among protein complexes decoys by challenging their thermal stability (*Radom et al., 2018*). All-atom MD simulations were carried out with pmemd within AMBER20 software suite for 50 representative models of CD4 NTD sCov2 RBD interaction obtained as described above. Glycosylation at sCov2 RBD N343 residue (*Watanabe et al., 2020*; *Woo et al., 2020*) was built using the Glycam web server. Each system was solvated in a periodic octahedral water box such that the initial structure was more than 24 Å of its closest image. Neutralizing ions ($Na^+$ and $Cl^-$) were added to a final concentration of 0.15 M (*Machado and Pantano, 2020*). The ff14SB force field (*Maier et al., 2015*) was used for modeling proteins, the GLYCAM06 force field (*Kirschner et al., 2008*) for modeling the glycan, and water was described with the TIP3P model (*Jorgensen et al., 1983*). Electrostatic interactions were evaluated using the Particle Mesh Ewald algorithm (*Darden et al., 1998*), nonbonded interactions were truncated at 9 Å, bonds involving hydrogen atoms were constrained at their equilibrium values, and a time step of 2 fs was used for the numerical integration of the equations of motion. Before production runs, all systems were equilibrated in five sequential steps: (1) 2500 steps of steepest descent line minimization followed by 2500 of conjugate gradient minimization; (2) 1 ns simulation at 303 K in the NVT ensemble with harmonic restraints of 10 kcal $mol^{-1}$ $Å^{-2}$ to protein and glycan atoms; (3) 1 ns simulation at 303 K in the NPT ensemble with harmonic restraints of 10 kcal $mol^{-1}$ $Å^{-2}$ to protein and glycan atoms; (4) 1 ns simulation at 303 K in the NPT ensemble with harmonic restraints of 10 kcal $mol^{-1}$ $Å^{-2}$ to protein backbone atoms; and (5) 1 ns simulation at 303 K in the NPT ensemble without restraints. Subsequently, production runs for assessment of the kinetic stability of the proposed structural models (*Radom et al., 2018*) were performed in the NPT ensemble in four sequential steps: (1) 50 ns simulation at 303 K; (2) 50 ns simulation at 323 K; (3) 50 ns simulation at 353 K; and (4) 50 ns simulation at 383 K. At each production step, models were evaluated according to their root-mean-square deviation (RMSD) to the initial model such that models with RMSD > 4 Å did not proceed to the next step. The 4 Å threshold was rationalized based on the fluctuations observed in the MD simulations on the sCov2 RBD and CD4 monomers independently. These simulations revealed that both monomers exhibited fluctuations ranging between 2 Å and 4 Å when evaluated up to 323 K (*Figure 2—figure supplement 1*). Additionally, for the data contained in *Figure 2—figure supplement 1*, triplicate production runs were performed for 100 ns at 303 K in the NPT ensemble. All trajectory analyses were performed with CPPTRAJ (*Roe and Cheatham, 2013*). Model structural visualization and image rendering were performed either in PyMOL (The PyMOL Molecular Graphics System, Version Schrödinger, LLC, n.d.) or VMD (*Humphrey et al., 1996*).

## Fluorescence anisotropy

Human CD4 (Sino Biological/10400-H08H), ACE2 (GenScript), and purified human thyroid receptor (TR) (negative control) proteins were labeled with FITC by incubation with fluorescein isothiocyanate probe, at molar ratio 3 FITC:1 protein, 4°C for 3 hr. The probe excess was removed by a desalting column (HiTrap 5 ml, GE) in a buffer containing 137 mM NaCl, 10 mM Na phosphate, 2.7 mM KCl, pH of 7.4. To evaluate the affinities between RBD and CD4, TR or ACE2, serial dilutions of RBD (0.01–10 µM) were performed. To evaluate the affinity between Spike full length (LECC/UFRJ LM220720) and CD4, serial dilutions of Spike (0.01–4.5 µM) were performed. The anisotropy curves were assembled in 0three replicates in black 384-well plates (Greiner). After serial dilution setup, each point was incubated with 20 nM of the labeled proteins (CD4, ACE2, or TR) for at least 2 hr at 4°C. For

each fluorescence curve, the mixtures were submitted to fluorescence anisotropy measurements using ClarioStar plate reader (BMG, using polarization filters of 520 nm for emission and of 495 nm for excitation). Data analysis was performed using OriginPro 8.6 software, and dissociation constant (Kd) was obtained from data fitted to binding curves through Hill model.

## Acknowledgements

We acknowledge the technical support of Elzira E Saviani, Paulo A Baldasso, and Mariana Ozello Baratti. We thank Dr. Leda Castilho (UFRJ) for providing sCoV-2 recombinant protein, Dr. Edson Luiz Durigon (USP) for providing viral strains, and Dr. Hernandes F de Carvalho (UNICAMP) for providing valuable reagents for immunofluorescence. We also thank the National Institute of Science and Technology of Photonics Applied to Cell Biology (INFABIC) for aid with confocal microscopy. We also thank Antonio Borges, Fabiano Montoro, and LNNano/CNPEM for the use of the electron microscopy facility (TEM-C2-26912, TEM-C2-26935). We acknowledge the Spectroscopy and Calorimetry Laboratory of the Brazilian Biosciences National Laboratory (LNBio), CNPEM, Campinas, Brazil, for the support in fluorescence anisotropy assays. This work was supported by grants from FAEPEX-UNICAMP (#2295/20, #2458/20, #2266/20 and #2274/20), São Paulo Research Foundation (FAPESP) (#2021/08354-2, #2019/16116-4, #2019/06372-3, #2020/04583-4, #2013/08293-7, #2020/04579-7, #2015/15626-8, #2018/14933-2, #2020/04746-0, #2019/00098-7, #2020/04919-2, #2017/01184-9, #2019/14465-1, #2020/04558-0, #2016/00194-8), the National Institute of Science and Technology in Neuroimmunomodulation (INCT-NIM) (#465489/2014-1) and FINEP (#01.20.0003.00). ASF and MAM. were supported by CNPq productivity awards (#306248/2017-4 and #310287/2018-9). AJRF, GGD, NSB, LBM, FC, VCC, AB, TLK, GSP, and RGL were supported by FAPESP fellowships (#2019/17007-4, #2016/18031-8, #2019/22398-2, #2019/13552-8, #2019/05155-9, #2019/06459-1, #2019/04726-2, #2017/23920-9, #2016/24163-4, #2016/23328-0). VOB and LNS were supported by FAEPEX fellowship (#2295/20 and #2319/20). DM was supported by CAPES fellowship.

## Additional information

### Funding

| Funder | Grant reference number | Author |
| --- | --- | --- |
| Fundo de Apoio ao Ensino, à Pesquisa e Extensão, Universidade Estadual de Campinas | #2295/20 | Vinicius O Boldrini<br>Luana Nunes Santos<br>Daniel Martins-de-Souza<br>Jose Luiz Proenca-Modena<br>Alessandro S Farias |
| Fundação de Amparo à Pesquisa do Estado de São Paulo | #2021/08354-2 | Marco AR Vinolo |
| Fundação de Amparo à Pesquisa do Estado de São Paulo | #2015/15626-8 | Helder I Nakaya |
| Fundação de Amparo à Pesquisa do Estado de São Paulo | #2019/14465-1 | Ana Carolina M Figueira |
| Instituto Nacional de Ciência e Tecnologia em Neuroimunomodulação | #465489/2014-1 | Alessandro S Farias |
| Financiadora de Estudos e Projetos | #01.20.0003.00 | Rafael E Marques |
| Coordenação de Aperfeiçoamento de Pessoal de Nível Superior | | Daniel Martins-de-Souza |

| Funder | Grant reference number | Author |
|---|---|---|
| Conselho Nacional de Desenvolvimento Científico e Tecnológico | #306248/2017-4 | Marcelo A Mori<br>Alessandro S Farias |
| Fundação de Amparo à Pesquisa do Estado de São Paulo | #2019/17007-4 | Natalia S Brunetti |
| Fundação de Amparo à Pesquisa do Estado de São Paulo | #2019/04726-2 | Thiago L Knittel |
| Fundo de Apoio ao Ensino, à Pesquisa e Extensão, Universidade Estadual de Campinas | #2319/20 | Vinicius O Boldrini<br>Luana Nunes Santos<br>Daniel Martins-de-Souza<br>Jose Luiz Proenca-Modena<br>Alessandro S Farias |
| Fundo de Apoio ao Ensino, à Pesquisa e Extensão, Universidade Estadual de Campinas | #2274/20 | Vinicius O Boldrini<br>Luana Nunes Santos<br>Daniel Martins-de-Souza<br>Jose Luiz Proenca-Modena<br>Alessandro S Farias |
| Fundo de Apoio ao Ensino, à Pesquisa e Extensão, Universidade Estadual de Campinas | #2266/20 | Vinicius O Boldrini<br>Luana Nunes Santos<br>Daniel Martins-de-Souza<br>Jose Luiz Proenca-Modena<br>Alessandro S Farias |
| Fundo de Apoio ao Ensino, à Pesquisa e Extensão, Universidade Estadual de Campinas | #2458/20 | Vinicius O Boldrini<br>Luana Nunes Santos<br>Daniel Martins-de-Souza<br>Jose Luiz Proenca-Modena<br>Alessandro S Farias |
| Fundação de Amparo à Pesquisa do Estado de São Paulo | #2019/16116-4 | Marco AR Vinolo |
| Fundação de Amparo à Pesquisa do Estado de São Paulo | #2019/06372-3 | Marco AR Vinolo |
| Fundação de Amparo à Pesquisa do Estado de São Paulo | #2020/04583-4 | Marco AR Vinolo |
| Fundação de Amparo à Pesquisa do Estado de São Paulo | #2013/08293-7 | Marco AR Vinolo |
| Fundação de Amparo à Pesquisa do Estado de São Paulo | #2020/04579-7 | Marco AR Vinolo |
| Fundação de Amparo à Pesquisa do Estado de São Paulo | #2018/14933-2 | Helder I Nakaya |
| Fundação de Amparo à Pesquisa do Estado de São Paulo | #2020/04746-0 | Helder I Nakaya |
| Fundação de Amparo à Pesquisa do Estado de São Paulo | #2019/00098-7 | Helder I Nakaya |
| Fundação de Amparo à Pesquisa do Estado de São Paulo | #2020/04919-2 | Helder I Nakaya |
| Fundação de Amparo à Pesquisa do Estado de São Paulo | #2017/01184-9 | Helder I Nakaya |

| Funder | Grant reference number | Author |
|---|---|---|
| Fundação de Amparo à Pesquisa do Estado de São Paulo | #2020/04558-0 | Ana Carolina M Figueira |
| Fundação de Amparo à Pesquisa do Estado de São Paulo | #2016/00194-8 | Ana Carolina M Figueira |
| Fundação de Amparo à Pesquisa do Estado de São Paulo | #2016/18031- 8 | Natalia S Brunetti |
| Fundação de Amparo à Pesquisa do Estado de São Paulo | #2019/22398-2 | Natalia S Brunetti |
| Fundação de Amparo à Pesquisa do Estado de São Paulo | #2019/13552-8 | Natalia S Brunetti |
| Fundação de Amparo à Pesquisa do Estado de São Paulo | #2019/05155-9 | Natalia S Brunetti |
| Fundação de Amparo à Pesquisa do Estado de São Paulo | #2019/06459-1 | Natalia S Brunetti |
| Fundação de Amparo à Pesquisa do Estado de São Paulo | #2017/23920-9 | Thiago L Knittel |
| Fundação de Amparo à Pesquisa do Estado de São Paulo | #2016/24163-4 | Thiago L Knittel |
| Fundação de Amparo à Pesquisa do Estado de São Paulo | #2016/23328-0 | Thiago L Knittel |
| Conselho Nacional de Desenvolvimento Científico e Tecnológico | #310287/2018-9 | Marcelo A Mori Alessandro S Farias |

The funders had no role in study design, data collection and interpretation, or the decision to submit the work for publication.

## Author contributions

Natalia S Brunetti, Validation, Investigation, Methodology, Writing – original draft, Writing – review and editing; Gustavo G Davanzo, Conceptualization, Data curation, Formal analysis, Supervision, Funding acquisition, Investigation, Methodology, Writing – original draft, Project administration, Writing – review and editing; Diogo de Moraes, Allan JR Ferrari, Data curation, Formal analysis, Investigation, Methodology, Writing – original draft; Gabriela F Souza, Lauar B Monteiro, João Victor Virgílio-da-Silva, Investigation, Methodology, Writing – original draft; Stéfanie Primon Muraro, Investigation, Visualization, Methodology; Thiago L Knittel, Validation, Investigation, Methodology, Writing – original draft; Vinicius O Boldrini, Gerson S Profeta, Natália S Wassano, Luana Nunes Santos, Flavio P Veras, Julia Forato, Mariene R Amorim, Fabiana Granja, Sean Whelan, André S Vieira, Investigation, Methodology; Victor C Carregari, Formal analysis, Investigation, Methodology; Artur HS Dias, Icaro MS Castro, Robson F Carvalho, Data curation, Formal analysis, Methodology; Lucas A Tavares, André C Palma, Eli Mansour, Raisa G Ulaf, Ana F Bernardes, Thyago A Nunes, Luciana C Ribeiro, Marcus V Agrela, Maria Luiza Moretti, Lucas I Buscaratti, Raissa G Ludwig, Jaqueline A Gerhardt, Natália Munhoz-Alves, Ana Maria Marques, Renata Sesti-Costa, Daniel A Toledo-Teixeira, Pierina Lorencini Parise, Matheus Cavalheiro Martini, Karina Bispos-dos-Santos, Camila L Simeoni, Murilo Carvalho, Bianca G Castelucci, Alexandre B Pereira, Laís D Coimbra, Patricia B Rodrigues, Arilson Bernardo SP Gomes, Fabricio B Pereira, Louis-Marie Bloyet, Spencer Stumpf, Marjorie C Pontelli, Andrei C Sposito, Methodology; Lícia C Silva-Costa, Fernanda Crunfli, Marieli MG Dias, Formal analysis, Methodology; Virgínia C Silvestrini, Eduardo B de Oliveira, Vitor M Faca, Leonilda MB Santos, Licio Velloso, Thiago

Mattar Cunha, Resources, Methodology; Marco AR Vinolo, Resources, Funding acquisition, Methodology, Writing – original draft; André Damasio, Luis LP da Silva, Resources, Investigation, Methodology; Ana Carolina M Figueira, Funding acquisition, Methodology; Helder I Nakaya, Resources, Data curation, Formal analysis, Funding acquisition, Methodology; Henrique Marques-Souza, Resources, Formal analysis, Methodology; Rafael E Marques, Resources, Formal analysis, Investigation; Daniel Martins-de-Souza, Resources, Formal analysis, Funding acquisition, Investigation, Methodology, Writing – original draft; Munir S Skaf, Conceptualization, Resources, Data curation, Formal analysis, Funding acquisition, Writing – original draft; Jose Luiz Proenca-Modena, Conceptualization, Resources, Supervision, Funding acquisition, Investigation, Methodology, Writing – original draft, Writing – review and editing; Pedro MM Moraes-Vieira, Marcelo A Mori, Alessandro S Farias, Conceptualization, Resources, Data curation, Formal analysis, Supervision, Funding acquisition, Writing – original draft, Project administration, Writing – review and editing

### Author ORCIDs
Natalia S Brunetti http://orcid.org/0000-0003-2803-3763
Allan JR Ferrari https://orcid.org/0000-0002-9199-2257
Stéfanie Primon Muraro https://orcid.org/0000-0002-5105-6659
Luana Nunes Santos http://orcid.org/0000-0003-0092-0803
Lucas A Tavares https://orcid.org/0000-0001-7883-1443
Eli Mansour http://orcid.org/0000-0001-6450-6930
Thyago A Nunes https://orcid.org/0000-0003-0202-0550
Natália Munhoz-Alves http://orcid.org/0000-0003-2756-3666
Daniel A Toledo-Teixeira https://orcid.org/0000-0002-4055-058X
Karina Bispos-dos-Santos http://orcid.org/0000-0003-0305-367X
Marieli MG Dias https://orcid.org/0000-0002-0246-8884
Louis-Marie Bloyet http://orcid.org/0000-0002-5648-3190
Robson F Carvalho https://orcid.org/0000-0002-4901-7714
Luis LP da Silva https://orcid.org/0000-0003-3558-0087
Thiago Mattar Cunha http://orcid.org/0000-0003-1084-0065
Helder I Nakaya http://orcid.org/0000-0001-5297-9108
Henrique Marques-Souza http://orcid.org/0000-0002-5008-8413
Jose Luiz Proenca-Modena http://orcid.org/0000-0002-4996-3153
Alessandro S Farias http://orcid.org/0000-0001-6759-1819

### Ethics

The samples from patients diagnosed with COVID-19 were obtained at the Clinical Hospital of the University of Campinas (SP-Brazil). Both COVID-19 patients and healthy subjects included in this work signed a term of consent approved by the University of Campinas Committee for Ethical Research (CAAE: 32078620.4.0000.5404, CAEE: 30227920.9.0000.5404). Human blood samples from severe COVID-19 patients analyzed in this study were obtained from individuals admitted at the Clinics Hospital, University of Campinas and included in a clinical trial (UTN: U1111-1250-1843). Besides, in vitro experiments were performed with buffy coats from healthy blood donors provided by The hematology and hemotherapy Center of the University of Campinas (CAAE: 31622420.0.0000.5404).

### Decision letter and Author response
Decision letter https://doi.org/10.7554/eLife.84790.sa1
Author response https://doi.org/10.7554/eLife.84790.sa2

## Additional files

### Supplementary files
• Supplementary file 1. Clinical data about cohorts. Gender, age, and admittance oxygen saturation were evaluated.

• Supplementary file 2. Detail results of mass spectrometry-based shotgun proteomics performed in CD4$^+$ T cells exposed to SARS-CoV-2 ex vivo.

• Supplementary file 3. Information about antibodies used in the article.

• Supplementary file 4. Cycle thresholds (Cts) obtained during qPCR performances in CD4$^+$ and CD8$^+$ T lymphocytes infection (*Figure 1A*) and CD4$^+$ T cell temporal infection (*Figure 1D*).

• MDAR checklist

## Data availability

Proteomic raw data are available in ProteomeXchange database under accession number PXD020967, Molecular dynamics dataset is available at https://github.com/ajrferrari/CD4-RBD-interaction-models, copy archived at *Ferrari, 2023*.

The following dataset was generated:

| Author(s) | Year | Dataset title | Dataset URL | Database and Identifier |
|---|---|---|---|---|
| Farias AS | 2023 | SARS-CoV-2 Uses CD4 to Infect T Helper Lymphocytes | https://www.ebi.ac.uk/pride/archive/projects/PXD020967 | PRIDE, PXD020967 |

The following previously published dataset was used:

| Author(s) | Year | Dataset title | Dataset URL | Database and Identifier |
|---|---|---|---|---|
| Liao M, Liu Y, Yuan J, Wen Y, Xu G, Zhao J, Cheng L, Li J, Wang X, Wang F, Liu L, Amit I, Zhang S, Zhang Z | 2020 | Single-cell landscape of bronchoalveolar immune cells in COVID-19 patients | https://www.ncbi.nlm.nih.gov/geo/query/acc.cgi?acc=GSE145926 | NCBI Gene Expression Omnibus, GSE145926 |

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
