## [Editor Report]

This study represents an important contribution where SARS-CoV-2 infection of T-helper cells is implicated and found to be mediated by CD4. It identifies the interaction between spike RBD domain and N Terminal domain of CD4 molecule as the specific viral attachment strategy. This solid paper also provides a potential usefulness for future work in understanding how viruses mediate infection of T cells.

---

## [Decision Letter]

**Decision letter after peer review:**

Thank you for submitting your article "SARS-CoV-2 Uses CD4 to Infect T Helper Lymphocytes" for consideration by *eLife*. Your article has been reviewed by 2 peer reviewers, and the evaluation has been overseen by a Reviewing Editor and Jos van der Meer as the Senior Editor.

Essential revisions:

Comments and suggestions:

1. A weakness of the paper is the reference citations.

Please be consistent in maintaining the citation style and numbering in the manuscript.

2. Although the extended Figure 9 suggests the expression of a multitude panel of gene expression, including apoptotic genes, the authors could not provide a piece of direct evidence to show how CD4 T cell death happens. What is the underlying mechanism of cell death? Is it by necrosis or apoptosis or pyroptosis? Please discuss.

3. Some previous studies suggested that lymphocytopenia in COVID infection could be due to impaired T cell proliferation or extravasation of T cells into tissue. The exact mechanism for lymphocytopenia should be discussed.

4. The data on CREB-1 Ser 133 in Figure 4E is not sufficiently convincing. It is difficult to understand what is the difference between every three lanes within mock and SARS CoV-2 infection. There is a pCREB band in lane 5 (2nd lane of CoV-2), but not in the other two. Please clarify.

5. Please add more specific information on the Insitu hybridization method. What is PK in the Insitu hybridization method? '…a couple of more washes'. Please be specific on experimental details.

6. Detection of antisense strand: Any reason to use 4% agarose gel? Is it due to the difficulty to detect the smaller size of the amplicon?

7. Immunoblotting: More experimental details are required. How many cells were infected or what is the total number of cells started with to get MOI=1? What is the exact time point used in Figure 4. E?

8. Immunofluorescence: Normally immunofluorescence protocol suggests 5% BSA for blocking. Is there any specific reason to use 1% BSA for blocking?

9. Transmission electron microscopy: More details on cacodylate buffer with calcium chloride in terms of concentration/molarity.

10. RNA extraction and cDNA synthesis: Please specify the sample volumes used for extracting the RNA. Specify the quantity of total RNA or cDNA used for qRT PCR.

11. What is the housekeeping gene HPRT? Please expand it.

12. Please watch the abbreviations and provide the expanded version wherever required. For example Single-cell mRNA seq: What is CCA? What is PCA?

13. It is interesting from a computational biology perspective that if GO database is removed (too unvalidated), and "Cell surface" component of the Jensen database (considering its more dedicated "Plasma membrane" and "External side of plasma membrane" components considered in the work) out of the Venn diagram (Extended Data Figure 3), then what will be left is more interaction partners shared between the remaining 3 databases. Interestingly, these additional partners would include CD8A and CD8B. However, the authors show that the interaction was experimentally noted to happen with CD4^+^ T cells but not with CD8^+^ ones. This warrants some discussion on why this might be the case. Moreover, what would be the computational docking/MD results if the authors were to attempt modelling an interaction between the spike glycoprotein and CD8? Would they not arrive at stable complexes with the MD workflow and 4 Å cutoff for temperature-induced stability scrutinization, that would be extra validation and weight on the adopted computational scheme for the discovery.

14. Looking at the last complex in Figure 2, where the full-length sCov2 is recovered on top of the modelled fragment, one can see some additional interaction points or potential clashes with CD4 NTD. Were some of the models discarded on the ground of the orientation between CD4 NTD and sCov2 RBD being incompatible with the full length sCov2 due to possible steric clashes?

15. The 4 Å cutoff for the temperature gradient-based structural stability check sounds reasonable, but would be more justifiable if the authors would also present a histogram of all RMSDs (of final aberrations) for all the tried models and show how outlying the 4 Å is in the whole distribution, additionally attributing a p-value on the selected cutoff.

16. "Methods" section, "Target selection" subsection. Please, define/describe LR in the text for the benefit of the readers.

17. "Methods" section, the last "Molecular Dynamics Simulations" subsection. There is a sentence that has "… for the data contained in Figure Sx…", please correct which supplementary figure is meant, as likely the Sx notation is left from the earlier drafts of the manuscript.

18. Figure 2 caption, an extra "." after (B).

19. Figure 2. It would help if model 95 and model 148 were brought roughly in the same spatial orientation as the final CD4 NTD and full sCov2 RBD model, to aid the readers in matching and getting more insight.

20. Extended Data Figure 3. In Figure A (Venn diagram), at "Plasma membrane Jensen compartments" notation closing bracket is missing.

---

## [Author Response]

Essential revisions:Comments and suggestions:1. A weakness of the paper is the reference citations.Please be consistent in maintaining the citation style and numbering in the manuscript.

We have edited the references according to the *eLife* citation style.

2. Although the extended Figure 9 suggests the expression of a multitude panel of gene expression, including apoptotic genes, the authors could not provide a piece of direct evidence to show how CD4 T cell death happens. What is the underlying mechanism of cell death? Is it by necrosis or apoptosis or pyroptosis? Please discuss.

We would like to thank the Reviewer for the question, which gave us the opportunity to review the text and include a discussion about this topic in the paper. Please see sentences between lines 240 and 246. SARS-CoV-2 can induce cell death by different mechanisms, depending on the infected cell type and the microenvironment and inflammatory condition where these cells are inserted. For example, SARS-CoV-2 can induce apoptosis in different cell lines through stimulation of autophagy via suppression of mTOR signaling (DOI: 10.1016/j.bbadis.2021.166260). SARS-CoV-2 can also induce inflammasome activation and cell death by pyroptosis in primary monocytes and PBMCs from COVID-19 patients (DOI: 10.1084/jem.20201707). Immunomodulatory drugs that inhibit inflammasome pathways impair SARS-CoV-2 replication in monocytes and other cell types (DOI: 10.1126/sciadv.abo5400) and may have beneficial effects to the treatment of COVID-19. In addition, SARS-CoV-2 drives multimodal necrosis in epithelial respiratory and kidney cell lines by a mechanism dependent on Open Reading Frame-3a of the virus (doi:10.1038/s41419‐018‐0917‐y) and it is able to induce PANoptosis by a mechanism dependent on ISG production, such as Z-DNA-binding protein 1 (ZBP1), in a context of cytokine storm and IFN-I treatment (10.1126/sciimmunol.abo6294). Thus, cell death induced by SARS-CoV-2 appears to play a pivotal role in the pathogenesis of COVID-19. Interestingly and in line with this idea, proinflammatory cytokines produced by SARS-CoV-2-infected apoptotic cells inhibit the expression of efferocytic receptors and impairs the continual process of clearance of dying cells by macrophages (DOI: 10.7554/*eLife*.74443). In the present paper, we demonstrated that SARS-CoV-2 can infect and induce cell death of CD4 T cells. Although the mechanisms of cell death have not been explored in depth in this article, data in the literature suggests that apoptosis is the mechanism underlying SARS-CoV-2-infected T cell death (DOI: 10.1093/jmcb/mjac021).

3. Some previous studies suggested that lymphocytopenia in COVID infection could be due to impaired T cell proliferation or extravasation of T cells into tissue. The exact mechanism for lymphocytopenia should be discussed.

Lymphocytopenia is a common characteristic of COVID-19 (https://www.nature.com/articles/s41392-020-0148-4). The decrease of the number of circulating lymphocytes might be due to innumerous factors, such as cell death, migration into the inflamed tissue as well as by promoting systemic inflammation and direct neutralization in human spleen and lymph nodes (https://doi.org/10.3389/fimmu.2021.661052). Single-cell analysis demonstrated that lymphocytes are not the main infiltrated cell population in the infected tissue (see figure 1E). Hence, it is likely that the decrease in lymphocyte count is attributed to both infected T cell death and dysfunctional T cells resulting from the inflammatory cytokine storm. This discussion has been incorporated into the manuscript.

4. The data on CREB-1 Ser 133 in Figure 4E is not sufficiently convincing. It is difficult to understand what is the difference between every three lanes within mock and SARS CoV-2 infection. There is a pCREB band in lane 5 (2nd lane of CoV-2), but not in the other two. Please clarify.

Each lane in the western blotting experiment represents a different sample and each group were done in triplicate (see Author response image 1). Hence, a combination of experimental and biological variability, which is common when using human samples, explains the differences between the lanes, particularly in the 5th lane of pCREB.

**Author response image 1. sa2fig1:** 

5. Please add more specific information on the Insitu hybridization method. What is PK in the Insitu hybridization method? '…a couple of more washes'. Please be specific on experimental details.

PK is an abbreviation for Proteinase K (https://www.sigmaaldrich.com/BR/pt/product/mm/124568) used in the antigen retrieval step by digesting mRNA bound proteins that could prevent the DNA probes to hybridize to their target mRNA. As for the wash steps after fixation with 4% PFA, three washes were performed to remove possible fixative residues. More specific information was added to the manuscript.

6. Detection of antisense strand: Any reason to use 4% agarose gel? Is it due to the difficulty to detect the smaller size of the amplicon?

The 4% agarose gel was employed in this protocol to attain the necessary resolution for discriminating between the sense and antisense strands based on their size difference. The difference in size between the two strands is only 21 nucleotides, which is caused by the non-viral tag present in the antisense strand. To effectively separate the small DNA fragments, a higher percentage agarose gel was required as it provides better resolution.

7. Immunoblotting: More experimental details are required. How many cells were infected or what is the total number of cells started with to get MOI=1? What is the exact time point used in Figure 4. E?

Most of the experiments were done with one million sorted CD3+CD4^+^ live cells plated over a well of a 48-well plate. For proteomic and immunoblotting experiments, four million live CD3+CD4^+^ cells were used. All experiments with SARS-CoV-2 were done in a biosafety level 3 laboratory (BSL-3) located in the Institute of Biology from the University Campinas. This information was clarified in the text.

8. Immunofluorescence: Normally immunofluorescence protocol suggests 5% BSA for blocking. Is there any specific reason to use 1% BSA for blocking?

Percentage of BSA for blocking in immunofluorescence usually varies from 1-5%. We used different concentrations and decided to follow the manufacturer's suggestion (https://www.abcam.com/protocols/immunocytochemistry-immunofluorescence-protocol).

9. Transmission electron microscopy: More details on cacodylate buffer with calcium chloride in terms of concentration/molarity.

We added the following sentence to the manuscript:

“T lymphocyte cell cultures were pelleted by centrifugation at 500 x g for 10 min. Cell culture supernatant was removed and cells were resuspended in 100 µL of the fixative solution, which consisted of 1 M sodium cacodylate aqueous solution supplemented with 2.5% v/v glutaraldehyde and 3 mM calcium chloride at pH 7.2, and kept overnight (16h) at 4°C”.

10. RNA extraction and cDNA synthesis: Please specify the sample volumes used for extracting the RNA. Specify the quantity of total RNA or cDNA used for qRT PCR.

We included more information in the text regarding the methods for RNA extraction and cDNA synthesis. Briefly, pelleted cells were resuspended in 200 µL of TRI Reagent and the RNA extraction performed following the manufacturer's instructions. 500 ng of RNA was converted into cDNA. The cDNA final concentration was 25 ng/μL and qPCR reaction was performed using 15 ng of cDNA.

11. What is the housekeeping gene HPRT? Please expand it.

The HPRT is a gene that encodes the hypoxanthine phosphoribosyltransferase 1 protein, an enzyme responsible for recycling purines (purine salvage pathway) in DNA molecules. This enzyme catalyzes conversion of hypoxanthine to inosine monophosphate (IMP) and guanine to guanosine monophosphate (GMP), when transferring phosphoribose from phosphoribosyl pyrophosphate (PRPP). This gene is present in all somatic cells and is considered a housekeeping gene because GTP is necessary for DNA synthesis and cellular energy. For lymphocytes, especially in non-activated conditions, HPRT has shown to be a reliable housekeeping gene (https://pubmed.ncbi.nlm.nih.gov/35645343/).

12. Please watch the abbreviations and provide the expanded version wherever required. For example Single-cell mRNA seq: What is CCA? What is PCA?

We thank the Reviewer for pointing this out. Abbreviations were defined as they appeared for the first time.

13. It is interesting from a computational biology perspective that if GO database is removed (too unvalidated), and "Cell surface" component of the Jensen database (considering its more dedicated "Plasma membrane" and "External side of plasma membrane" components considered in the work) out of the Venn diagram (Extended Data Figure 3), then what will be left is more interaction partners shared between the remaining 3 databases. Interestingly, these additional partners would include CD8A and CD8B. However, the authors show that the interaction was experimentally noted to happen with CD4^+^ T cells but not with CD8^+^ ones. This warrants some discussion on why this might be the case. Moreover, what would be the computational docking/MD results if the authors were to attempt modelling an interaction between the spike glycoprotein and CD8? Would they not arrive at stable complexes with the MD workflow and 4 Å cutoff for temperature-induced stability scrutinization, that would be extra validation and weight on the adopted computational scheme for the discovery.

Although the Reviewer raises an interesting observation, our experiments beyond the computational prediction do not support a role for CD8 in T lymphocyte infection. Hence, we decided not to focus on a potential interaction between CD8 and SARS-CoV spike protein, although this does not exclude that the interaction may exist. Indeed, a highly stable interaction between Spike and CD8 molecules could result in a 'stable' complex that would trap the viral particles outside the cell precluding infection. This would be consistent with T CD8 cells not being infected by SARS-CoV-2. However, we would need more experiments to prove this mechanism and this is beyond the scope of this manuscript, which focuses on the interaction between spike and CD4 and its role in CD4 T cell infection.

14. Looking at the last complex in Figure 2, where the full-length sCov2 is recovered on top of the modelled fragment, one can see some additional interaction points or potential clashes with CD4 NTD. Were some of the models discarded on the ground of the orientation between CD4 NTD and sCov2 RBD being incompatible with the full length sCov2 due to possible steric clashes?

We appreciate the Reviewer's observation regarding the potential clashes between CD4 NTD and the full-length sCov2 in the last complex presented in Figure 2. In constructing the full model, we initially performed a superposition of the RBD domain onto the full-length sCov2. The primary criterion for selecting the final models was the kinetic stability analysis, as we aimed to identify the most stable configurations. We acknowledge that the compatibility between the docking model and the full-length sCov2, considering possible steric clashes, is an essential factor to consider. For the two most stable models, namely model95 and model148, we carefully examined their structures. In the case of model148, illustrated in Figure 2D, we did not observe any apparent clashes between CD4 NTD and the full-length sCov2.

However, we would like to emphasize that in model148, there is a possibility of CD4 NTD interacting with a second sCov2 RBD in its closed state. These interactions may involve polar interactions or mediation by water molecules when solvated. Nevertheless, based on our analysis, we do not anticipate that these few interactions play a significant role in the overall stability and function of the complex. To provide further clarity, we have made available a PyMOL session containing the structures used for the analysis in Figure 2. This session can be accessed via the following GitHub page: https://github.com/ajrferrari/CD4-RBD-interaction-models under the 'pymol-session' folder. We hope this additional information addresses the concern raised by the Reviewers.

15. The 4 Å cutoff for the temperature gradient-based structural stability check sounds reasonable, but would be more justifiable if the authors would also present a histogram of all RMSDs (of final aberrations) for all the tried models and show how outlying the 4 Å is in the whole distribution, additionally attributing a p-value on the selected cutoff.

We appreciate the Reviewer’s feedback regarding the justification for the 4 Å cutoff used in our temperature gradient-based structural stability check. We agree that providing additional analysis to support the chosen cutoff would enhance the validity of our findings. To further rationalize the selected cutoff, we conducted MD simulations on the sCov2 RBD and CD4 monomers independently (see figure 2—figure supplement 1e). These simulations revealed that both monomers exhibited fluctuations ranging between 2 Å and 4 Å when evaluated up to 323K. These fluctuations serve as a baseline, indicating the inherent flexibility and dynamics of the individual monomers. In our analysis, models that crossed the 4 Å cutoff at any point during the MD trajectory were considered kinetically unstable, as they exhibited higher fluctuations than the individual monomers. We have now clarified this rationale in the main text to provide a clearer understanding of the criteria used for assessing stability. To address the Reviewer’s suggestion, we have generated a histogram of the maximum backbone RMSD values for all the tried models. This histogram is provided in figure 2—figure supplement 1f.

It is important to note that the p-value obtained from the distribution of maximum RMSDs is 0.15. However, it is crucial to consider the characteristics of our evaluation process. Not all models completed the four MD steps, as some diffused away early during the evaluation. As a result, the distribution of maximum RMSDs observed is biased toward lower values. Consequently, the 4 Å cutoff appears less of an outlier than it would be if all models were equally sampled.

We believe that this additional analysis, including the histogram of maximum RMSDs, provides a more comprehensive understanding of the distribution and relative position of the 4 Å cutoff. It accounts for the limitations of the sampling process and strengthens the justification for its selection. We have revised the manuscript to include these findings, addressing the Reviewer’s' concern.

16. "Methods" section, "Target selection" subsection. Please, define/describe LR in the text for the benefit of the readers.

LR (likelihood ratio) has been defined in the manuscript.

17. "Methods" section, the last "Molecular Dynamics Simulations" subsection. There is a sentence that has "… for the data contained in Figure Sx…", please correct which supplementary figure is meant, as likely the Sx notation is left from the earlier drafts of the manuscript.

We have solved this issue in the new version of the manuscript.

18. Figure 2 caption, an extra "." after (B).

We have solved this issue in the new version of the manuscript.

19. Figure 2. It would help if model 95 and model 148 were brought roughly in the same spatial orientation as the final CD4 NTD and full sCov2 RBD model, to aid the readers in matching and getting more insight.

We appreciate the Reviewer's suggestion regarding Figure 2 and the importance of presenting model 95 and model 148 in a spatial orientation similar to that of the final CD4 NTD and full sCov2 RBD model. We agree that this would facilitate the readers' ability to match and gain deeper insights from the figure. In response to this valuable feedback, we have revised Figure 2 to ensure that model 95 and model 148 are now presented in a comparable spatial orientation to the final CD4 NTD and full sCov2 RBD model. This modification allows for a more straightforward visual comparison and enhances the clarity and comprehensibility of the figure. We also provide a pymol session that will allow the readers to explore in more details the models shown in Figure 2. The pymol session can be accessed via the following GitHub page: https://github.com/ajrferrari/CD4-RBD-interaction-models under the 'pymol-session' folder.

20. Extended Data Figure 3. In Figure A (Venn diagram), at "Plasma membrane Jensen compartments" notation closing bracket is missing.

We have solved this issue in the new version of the manuscript.